# Know When to Fold 'Em: Predicting an LLM-Judge for Efficient but Performant Inference

## Abstract

Large language models (LLMs) face a fundamental trade-off between computational efficiency (e.g., number of parameters) and output quality, especially when deployed on computationally limited devices such as phones or laptops. One way to address this challenge is by following the example of humans and have models ask for help when they believe they are incapable of solving a problem on their own; we can overcome this trade-off by allowing smaller models to respond to queries when they believe they can provide good responses, and deferring to larger models when they do not believe they can. To this end, in this paper, we investigate whether models can predict—prior to responding—how an LLM judge would score their output. We evaluate three approaches: zero-shot prediction, prediction using an in-context report card, and supervised fine-tuning. Our results show that larger models (particularly reasoning models) demonstrate good zero-shot prediction abilities, while smaller models require in-context report cards or fine-tuning for reliable predictions. While the effectiveness varies across datasets, both approaches can substantially improve smaller models' prediction accuracy, with fine-tuning achieving mean improvements up to 52% across datasets. These findings suggest that models can learn to predict their own performance limitations, paving the way for more efficient and self-aware AI systems.

## 1 Introduction

The rise of large language models (LLMs) has been the defining narrative of the recent AI boom. The increasing investment in AI brought about by this has led to increasingly large resources being pooled into training LLMs. The compounding effect of this has resulted in mainstream LLMs needing upward of a terabyte of GPU memory to run. Such models are both incredibly costly to use (see, e.g., Noffsinger et al. (2024)) and necessitate that consumer devices rely on API calls to access them—which users do, making more than 2.5 billion requests to ChatGPT a day (Roth, 2025). To address these limitations of large models, there has been growing interest in developing intentionally small models (e.g., Chiang et al. (2023); AI (2024); Project Apertus (2025)).

Small models, such as the recent MobileLLM (Liu et al., 2024b), demonstrate comparable performance to much bigger models on many datasets. However, they tend to demonstrate a much larger drop in their performance on other tasks compared with their bigger siblings (Pecher et al., 2022). This is particularly concerning as this performance drop is not uniform, leading to famously erratic responses such as models incorrectly responding to "how many Rs are there in strawberry?"(BlakeSergin, 2024). As such, it would be ideal to predict the quality of a response before the model actually responds, allowing us to reduce the likelihood of an erratic response without needing to restrict ourselves to larger models.

To evaluate the output of a model without constraining the kinds of outputs that can be given, one needs to employ something like an LLM-based (or agentic) judge (Zheng et al., 2023; Zhuge et al., 2024). However, this is costly as generating output tokens is dramatically more expensive than processing input tokens (Shi et al., 2024; Li et al., 2025a). This cost is both an increase in the literal cost and a delay to the user (as most responses are served to the user in a streaming manner). Additionally, as our results show, LLM judges exhibit a subpar evaluation of responses when doing so in isolation (compare Figures 8 to 12 with Figures 16 to 20 in Appendix E).

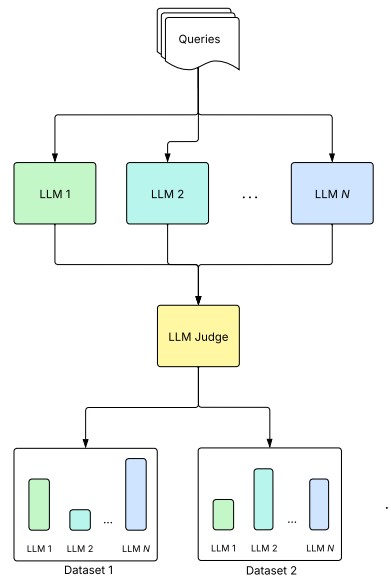

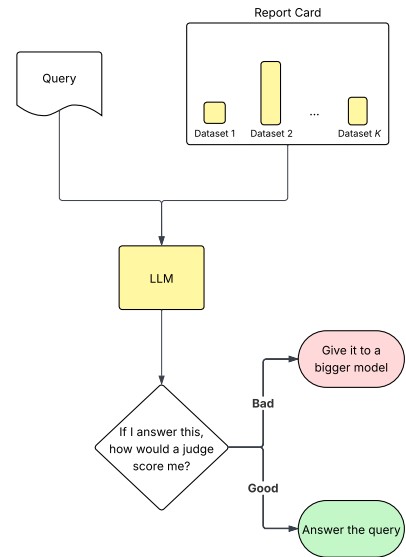

Figure 1: Report cards are generated by having a judge compare the responses of several LLMs over a number of datasets. This joint evaluation increases the diversity of responses by the judge (see Appendix E).

Figure 2: With a report card, a model is able to estimate its own capability to respond to a query, without falling prey to the characteristic overconfidence of LLMs. This can be used to inexpensively route queries.

This work investigates the ability for models to predict their own performance pre-hoc as evaluated by an LLM judge. We experiment with three approaches: tabula rasa (zero-shot), in-context learning, and fine-tuning to enable models to predict how an LLM-based judge (Zheng et al., 2023; Zhuge et al., 2024) would score their responses to queries. LLM judges offer several advantages over traditional evaluation metrics: they provide reliable evaluations that correlate strongly with human evaluations (Zheng et al., 2023; Wang et al., 2024b; Liu et al., 2023; Fu et al., 2024a; Zhuge et al., 2024), support flexible evaluation criteria, and can be easily adapted to new alignment problems by modifying their prompts (Zhuge et al., 2024).

We evaluate our approaches across models of varying sizes: MobileLLM 0.9B, Llama 3.1 8B, Llama 3.2 1B and 3B, Llama 3.3 70B, GPT OSS 20B and 120B, DeepSeek Distilled Qwen 14B and 32B, DeepSeek Distilled 70B, and Llama 4 Scout. Our findings reveal that while large models—particularly reasoning models—show a good ability to predict judge scores, smaller models typically suffer from miscalibration, being either overconfident or underconfident in their self-assessments. This miscalibration reflects a well-documented tendency of LLMs toward overconfidence (Huang et al., 2025; Ren et al., 2023).

To address these calibration issues, we propose two distinct approaches. Our first approach provides models with a *report card*—a detailed performance summary based on the model's historical performance across multiple datasets. This method requires no additional training and can be applied to any model, including closed-weight systems. We generate these report cards by evaluating models alongside other systems across diverse datasets (see Figure 1) and using the modal judge ratings to construct textual performance descriptions.

Our second approach fine-tunes models specifically for performance prediction. While this requires additional training, it offers greater inference efficiency by eliminating the need to process report card tokens. We construct training data using the hindsight trick (Andrychowicz et al., 2017), re-labeling examples with judge scores, then apply supervised fine-tuning (Ouyang et al., 2022) to several model variants including MobileLLM 0.9B, Llama 3.1 8B, and Llama 3.2 1B and 3B.

Both approaches successfully improve smaller models' ability to predict their own performance, with effectiveness varying significantly across datasets. Interestingly, models demonstrate stronger

self-awareness on more challenging queries, suggesting that difficulty may serve as a useful signal for performance prediction. These findings open promising directions for developing more reliable and self-aware small language models.

Our primary contributions can thus be summarized as that **(1)** we propose predicting how well LLMs are able to estimate pre-hoc how their response to a query would be scored by an LLM judge (thereby allowing flexible evaluations) and show that large LLMs demonstrate a fair ability to do so tabula rasa; **(2)** we propose using a report card system that summarizes the mode of the performance of a model in a number of datasets and demonstrate that it can dramatically improve the accuracy of many small models in the aforementioned prediction task; and **(3)** we propose bypassing the inference cost of the report card system by fine-tuning models using the hindsight trick, demonstrating that this leads to strong prediction performance in many small models.

## 2 PRELIMINARIES

### 2.1 LARGE LANGUAGE MODELS (LLMs)

LLMs are a class of (typically autoregressive) extremely large pretrained sequential models (in the billions of parameters), predominantly based on the transformer architecture (Vaswani et al., 2017) (see also the earlier Unnormalized Linear Transformer (Schmidhuber, 1992; Schlag et al., 2021)). They are trained on large text corpora, learning a probability distribution over a sequence of tokens which can be later used for text completion (i.e., text generation). As such, these models excel at understanding, generating, and processing human language. Mathematically, an LLM generates a sequence of tokens $y_1, \ldots, y_L$ for a given input prompt $x$ by modeling the conditional probability

$$P(y_1, \ldots, y_L | x; \theta) = \prod_{t=1}^{L} P(y_t | y_{<t}, x; \theta) \tag{1}$$

where $y_{<t}$ denotes previously generated tokens and $\theta$ represents the model's learned parameters. The transformer architecture (the backbone of most LLMs) models a sequence of input embeddings $X = (x_1, \ldots, x_n)$ using stacked layers of self-attention and feed-forward blocks. Each self-attention block computes new representations $Z$ as

$$Z = \text{softmax}\left(\frac{QK^\top}{\sqrt{d_k}}\right) V, \tag{2}$$

where $Q = XW_Q$, $K = XW_K$, $V = XW_V$ are learned linear projections of the input, and $d_k$ is the key dimension. Positional encodings are added to $X$ to retain order information. This architecture is exceptionally parallelizable and has an extraordinary ability for modeling long-range dependencies. Research has demonstrated that LLMs, particularly when augmented with an external read-write memory, are Turing complete, capable of simulating any algorithm (Schuurmans, 2023). The behavior of an LLM is heavily guided by *prompts*, with carefully crafted, context-rich instructions being essential for achieving high-quality and task-aligned outputs (FAANG, 2024). Although their pre-training imbues them with broad knowledge, to further align their output to human preferences and intents, LLMs are frequently fine-tuned (DeepSeek-AI et al., 2025; Muldrew et al., 2024; Sieker et al., 2024). This alignment helps to make models more helpful, harmless, and honest.

### 2.2 AGENTIC SYSTEMS, ALIAS AGENTIC AI

Building upon the capabilities of individual LLMs, *agentic systems* (sometimes called Agentic AI) are modular AI architectures that orchestrate one or more LLM invocations through arbitrary control flow, often integrating external tool calls (Schick et al., 2023; Zhuge et al.). These systems differ from standalone LLMs by their ability to engage in multi-step reasoning, planning, and acting to solve complex tasks autonomously or semi-autonomously (Zhuge et al.; 2024). This multi-step approach enables decomposition of complex problems into manageable subtasks and allows for iterative refinement based on intermediate results-capabilities that single-pass LLM inference cannot achieve (Zhuge et al., 2024). The internal workflow of an agentic system can be conceptualized as a directed graph $G = (V, E)$, where each node $v \in V$ represents an LLM call, tool execution, or decision point, and the edges $(u, v) \in E$ denote dependencies between actions (Zhuge et al., 2024).

| Shorthand | Full Name | # Parameters | Reference |
|---|---|---|---|
| M09B | MobileLLM 0.9B | 0.9 Billion | Liu et al. (2024b) |
| L318B | Llama 3.1 8B Instruct | 8.03 Billion | Grattafiori et al. (2024) |
| L321B | Llama 3.2 1B Instruct | 1.24 Billion | AI (2024) |
| L323B | Llama 3.2 3B Instruct | 3.21 Billion | AI (2024) |
| L3370B | Llama 3.3 70B Instruct | 70.6 Billion | AI (2024) |
| L416E | Llama 4 Scout 17B 16E Instruct | 109 Billion | AI (2025) |
| DSQ14B | DeepSeek R1 Distilled Qwen 14B | 14.8 Billion | DeepSeek-AI et al. (2025) |
| DSQ32B | DeepSeek R1 Distilled Qwen 32B | 32.8 Billion | DeepSeek-AI et al. (2025) |
| DSL70B | DeepSeek R1 Distilled Llama 70B | 70.6 Billion | DeepSeek-AI et al. (2025) |
| GPT20B | GPT OSS 20B | 21.5 Billion | OpenAI (2025) |
| GPT120B | GPT OSS 120B | 120 Billion | OpenAI (2025) |

Table 1: Models considered for our experiments. The parameter count is taken from HuggingFace Safetensor values to ensure equal treatment between models.

Tool use represents a fundamental form of agentic AI, enabling systems to dynamically interact with environments and access specialized capabilities such as code interpreters and search engines (Schick et al., 2023; Qin et al., 2024). This approach extends LLM capabilities beyond their training limitations through external resource integration.

A prominent application of agentic systems is automated evaluation through *LLM-as-a-Judge* frameworks, which utilize LLMs to assess text generation quality and model behavior (Li et al., 2024b). These frameworks can be extended into *Agent-as-a-Judge* systems that leverage full agentic capabilities to provide richer evaluation feedback, including assessment of tool use and multi-step reasoning processes (Zhuge et al., 2024).

## 3 EXPERIMENTAL SETUP

We experiment with five (5) datasets that are commonly used in the literature: MedQA, which contains queries asking to diagnose a medical condition (Jin et al., 2020); Long-Fact, which contains queries asking factual trivia (Wei et al., 2024); AIME 2024, which contains competition-level maths problems (HuggingFaceH4, 2025); SciCode, which contains requests for producing code for scientific purposes (Tian et al., 2024); and MMLU-Pro, which contains general undergraduate-level examination questions (Wang et al., 2024c). Altogether, these datasets cover a broad range of application cases for LLMs.

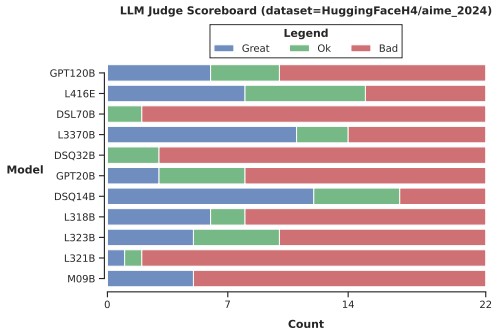

Figure 3: The distribution of the LLM judge scores for each of the models on the AIME 2024 dataset. Note that the poor performance of some reasoning models here was due to us limiting reasoning models to producing no more than 24576 tokens.

In total, we experimented with the eleven (11) models shown in Table 1. We use Llama 3.3 70B as our judge, instructing it to evaluate all the responses of the models to the query (we also provided any relevant answers included in the dataset to the judge) according to a predefined rubric. Llama 3.3 70B remains a well-studied model with robust performance, making it well-suited to use as a judge here. We include an ablation using GPT OSS 120B as a judge in Appendix G. The rubric used is shown in Prompt 11 alongside all the other prompts used in Appendix C. To ground evaluations more, we had the judge evaluate simultaneously the responses of all the models for one query simultaneously. We performed an ablation experiment that confirms the independent evaluations led to less diversity between the models in Appendix E. In addition to providing a grade for each response of either "great," "ok," or "bad," the judge was instructed to provide a rationale for why it gave each response the grade it gave them. We selected

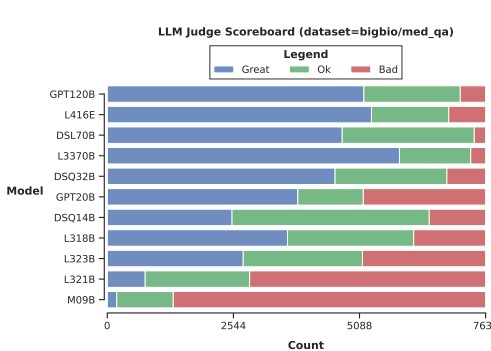

Figure 4: The distribution of the LLM judge scores for each of the models on the MedQA dataset. Note the wide spread of performance here across the different models.

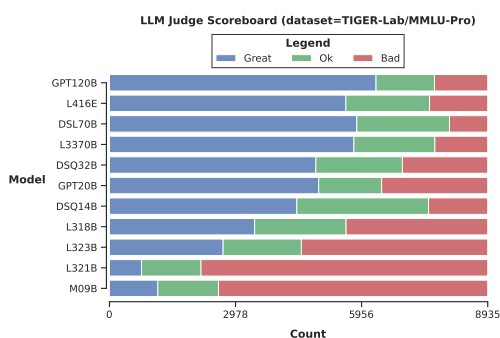

Figure 6: The distribution of the LLM judge scores for each of the models on the MMLU-Pro dataset. We provide a per-category breakdown in Figure 14 in Subsection 4.1.

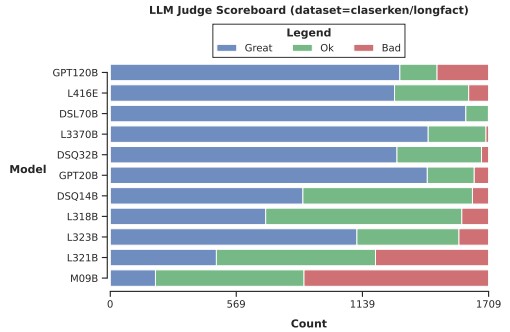

Figure 5: The distribution of the LLM judge scores for each of the models on the Long-Fact dataset. Much of this information is in-distribution, so most LLMs can do well here.

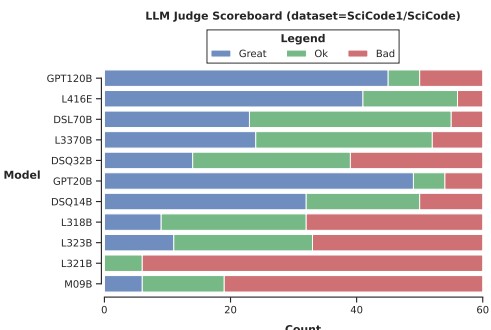

Figure 7: The distribution of the LLM judge scores for each of the models on the SciCode dataset. This dataset involves coding so most models struggle here.

Llama 3.3 70B as our ad-hoc observations found that it was able to effectively follow the instructions given and generally gave good rationales for its grading. In cases where the response given by the judge could not be parsed, one request was made to correct it. Cases where the corrected response could not be parsed were minimal.

The distribution of scores given by the LLM judge for each of the models on each of the datasets is shown in Figures 3 to 7. These results roughly conform to expectations, with most models doing (**1**) well with basic factual information, (**2**) poorly with mathematic questions, (**3**) larger and reasoning models doing better than smaller and non-reasoning models, and (**4**) newer models doing better for their size than older models.

## 4 ZERO-SHOT AND IN CONTEXT PREDICTION

LLMs have demonstrated strong generalization abilities across a broad range of tasks (Brown et al., 2020; Wei et al., 2022), making it reasonable to think that a model might be able to predict their performance pre-hoc. We experiment with the model's given in Table 1 ability to directly predict how an LLM judge would score their response without any context apart from the query (see Prompt 3 in Appendix C for the exact query given to the model here). The context for all the models was limited to one round of interactions, i.e., one query. The prediction performance of the models in the zero-shot case is shown alongside our other results in Figures 8 to 12. We provide a summary of these results in Table 2 in Appendix B.

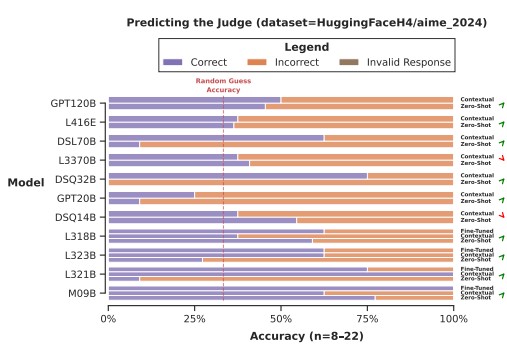

Figure 8: The zero-shot and contextual prediction accuracy of all the models on the AIME 2024 dataset (a green arrow indicates improvement over zero-shot). Note the dramatic improvement for smallest models here.

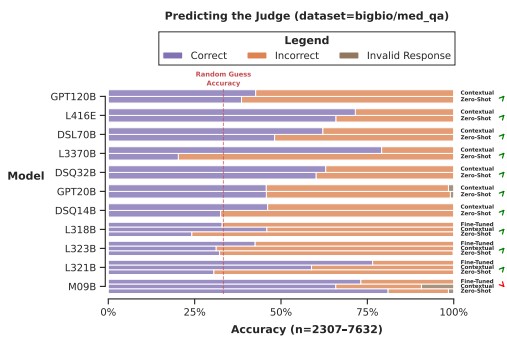

Figure 9: The zero-shot and contextual prediction accuracy of all the models on the MedQA dataset (a green arrow indicates improvement over zero-shot). Note how dramatic the improvement offered to even big non-reasoning models is (i.e., L3370B here).

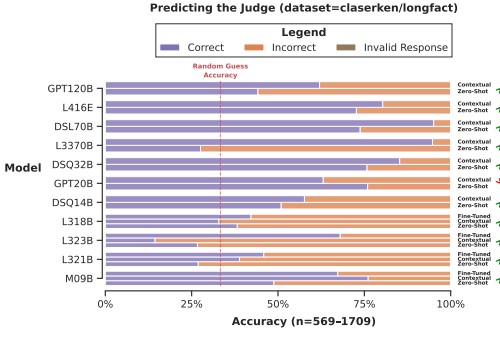

Figure 10: The zero-shot and contextual prediction accuracy of all the models on the Long-Fact dataset (a green arrow indicates improvement over zero-shot). Note that, even on a dataset where all the models do well, their prediction quality is better in the contextual/fine-tuning setting.

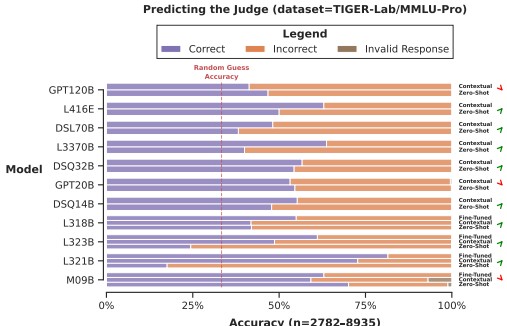

Figure 11: The zero-shot and contextual prediction accuracy of all the models on the MMLU-Pro dataset (a green arrow indicates improvement over zero-shot). We provide a per-category breakdown of the zero-shot results in Figure 15 in Section 4.1.

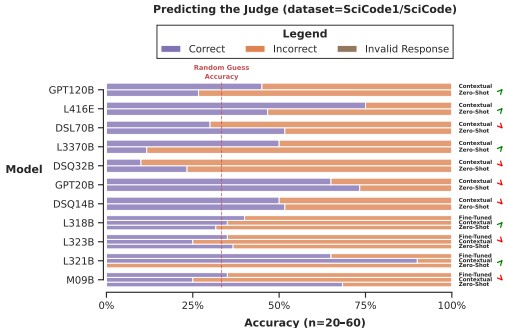

Figure 12: The zero-shot and contextual prediction accuracy of all the models on the SciCode dataset (a green arrow indicates improvement over zero-shot). This is the only dataset where the contextual/fine-tuning approach frequently didn't help. We hypothesize that the judge was not sophisticated enough to evaluate models on this dataset (see, e.g., Zhuge et al. (2024)).

| Dataset | M09B | L321B | L323B | L318B |
|---------|------|-------|-------|-------|
| MedQA | -.09 | +.46 | +.10 | +.09 |
| LongFact | +.19 | +.19 | +.41 | +.04 |
| AIME 2024 | +.23 | +.66 | +.35 | +.03 |
| MMLU-Pro | -.07 | +.64 | +.37 | +.13 |
| SciCode | -.33 | +.65 | -.02 | +.08 |
| *Mean* | -.02 | **+.52** | +.24 | +.07 |

Figure 13: Summary of the improvement in the prediction accuracy for the fine-tuned models as compared with the zero-shot setting. For a summary of the contextual setting results, see Table 2 in Appendix B.

The key observation to take from the zero-shot results in Figures 8 to 12 is that **(1)** the prediction performance of the models varied wildly based on the dataset used, **(2)** reasoning models exhibited an often better ability to estimate their own performance, and **(3)** smaller models typically performed at or worse than a random guess accuracy.

Based on the large variation observed in model and prediction performance by dataset, it seems reasonable that a promising approach would be to determine the kind of query and predict based on that. To that end, we produce a "report card" for each model based on their performance on the datasets (see Figures 3 to 7). This report card simply provides the mode of the scores for that model on that dataset. These mode scores are given in Appendix D with the report card structure shown in Prompt 5 in Appendix C (we provide an ablation on the report card structure in Prompt 6).

The results of the report card-based, or contextual prediction performance are given alongside the previous results in Figures 8 to 12. The effect of the report card is particularly good on the smaller and non-reasoning models. This suggests that the inclusion of the report card was effective with the small models on allowing them to categorize the query and match it to their skill sets, validating the hypothesis that even a model can predict its own performance with the right information.

## 4.1 MMLU-Pro Category Results

One concern with the LLM judge, is that the performance of it might be highly stochastic. To test this, we split the queries in the MMLU-Pro dataset up by category and observed the performance score distribution of the models and their predictions as a function of the category. Figure 14 shows the probability that the judge gives a score of "great" or "ok" for each model on the MMLU-Pro dataset as a function of the category. The conclusion that can be drawn from Figure 14 is that the distribution of scores given by the judge is much less affected by the category than by the model, implying a comparatively low degree of variance in the judging.

Figure 15 shows the probability that the model correctly predicts its score zero-shot. Again, the variance is more visible across models than categories, suggesting that the categorization ability of the models has not been deeply affected by the variance of the judge (note that we would expect roughly equal performance on a dataset given the homogeneity of the datasets in the kind of queries they ask and the broad generalization of the models in certain kinds of queries).

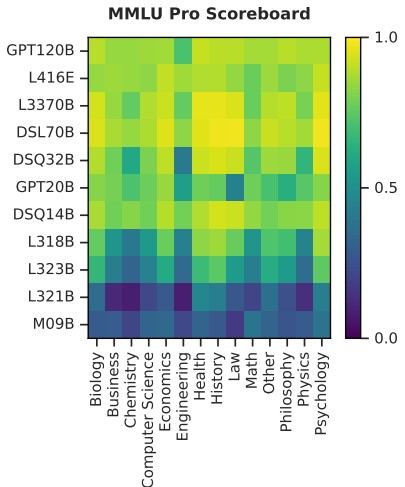
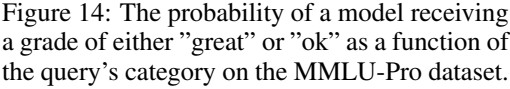
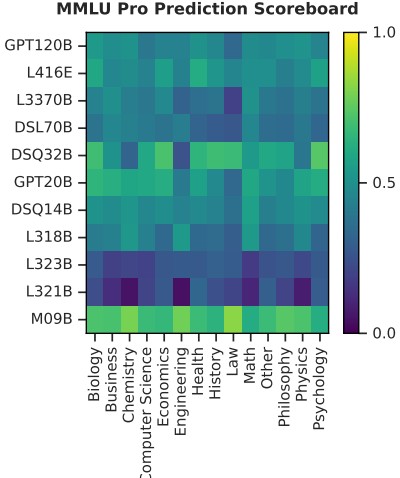

Figure 14: The probability of a model receiving a grade of either "great" or "ok" as a function of the query's category on the MMLU-Pro dataset.

Figure 15: The probability of a model correctly predicting its score zero-shot as a function of the query's category on the MMLU-Pro dataset.

## 5 FINE-TUNING APPROACH

The biggest drawback of using report cards to allow a model to estimate its performance on a query is that it necessitates a model processing the large number of tokens needed for a comprehensive report card. We conducted an ablation where we use shorter report cards (see Prompt 6), but this impacted performance. To get the best of both worlds, in this section we experimented with fine-tuning versions of the small models to perform these predictions without a report card.

To accomplish the above, we produced a dataset of zero-shot prompts and the respective zero-shot model evaluations collected in Section 4. We then applied the hindsight trick (see, e.g., Andrychow-icz et al. (2017)) to relabel the zero-shot predictions as though they were accurate. We then applied supervised fine-tuning (Ouyang et al., 2022) using this data this to MobileLLM 0.9B, Llama 3.1 8B, and Llama 3.2 1B and 3B. We refer to these fine-tuned models as MP09B, LP318, LP321B, and LP323B, respectively. Note that none of these models are reasoning models and so the respective output for the prediction should be one token. The hyperparameters used in this are available in our released source code (see Section 1).

The results of this fine-tuning approach are presented alongside our other results in Figures 8 to 12, with an arrow annotation indicating if either the contextual or fine-tuning approach improved the prediction performance. We provide a summary of these results in Figure 13. The prediction quality of the fine-tuned models is clearly at or above the performance of even the report card-based predictions. However, as with the report cards, the prediction quality is heavily dependent on the dataset and is much stronger in the smaller models (where the distribution of judge evaluations is more concentrated).

Altogether, this suggests that a fine-tuning–based approach is able to enable a model to predict its own performance at or above the endowment given by a report card.

## 6 RELATED WORK

**LLM and Agentic Judges.** The advent of Large Language Models (LLMs) has catalyzed their adoption as automated evaluators, particularly through the *LLM-as-a-Judge* paradigm (Zheng et al., 2023; Liu et al., 2023; Fu et al., 2024a). This approach leverages powerful foundation models to assess text quality, dialogue performance, and agent behavior, offering a scalable and cost-effective alternative to human evaluation. Studies demonstrate that single-LLM judges can achieve high correlation (Spearman coefficients of 0.7–0.9) with aggregated human preferences across diverse tasks (Zheng et al., 2023).

However, this paradigm faces big limitations. Single-judge systems are prone to systematic biases, including preferences for specific output lengths, styles, or verbosity (Dubois et al., 2024). They also exhibit vulnerabilities to adversarial attacks and may fail to detect nuanced factual inaccuracies or complex reasoning errors (Eiras et al., 2025; Li et al., 2025b). Recent work has further highlighted position bias (Wang et al., 2024a) and limited reasoning depth in complex evaluation scenarios.

To address these challenges, the research community has progressed to *Agent-as-a-Judge* and multi-agent evaluation frameworks. These systems employ multiple LLM agents that collaborate, debate, or assume specialized roles (e.g., Grader, Critic, Defender) to produce more robust and reliable assessments (Zhuge et al., 2024; Chen et al., 2025). Notable implementations include MAJ-EVAL (Chen et al., 2025), featuring a multi-agent jury; AGENTSCOURT (He et al., 2024) for competitive agent environments; and FINCON (Yu et al., 2024) for financial reasoning evaluation. Recent advancements also explore *debate-based evaluation* (Chan et al., 2024) and *iterative refinement* processes (Liang et al., 2024) to enhance judgment quality.

Nevertheless, multi-agent judges introduce new complexities, such as potential collusion when agents share similar model backbones, emergent group biases, and increased computational costs (Chan et al., 2024; Liang et al., 2024). The search for optimal agent architectures and interaction protocols remains an active area of research (Qin et al., 2024; Wu et al., 2024).

**Our work diverges** from these directions by not aiming to improve the judging mechanism itself. Instead, we focus on *preemptively predicting the scores* that an LLM or agentic judge would assign to a model's response *before* its generation. This approach enables more efficient system design,

rapid prototyping, and optimal resource allocation by shifting the focus from *being* the evaluator to *anticipating* the evaluation outcome.

**Predicting Model Performance.** Another line of research explores models' abilities to self-assess or predict aspects of their own performance, particularly regarding confidence estimation and hallucination mitigation. CONFQA, for example, introduces a fine-tuning strategy that explicitly trains LLMs to express uncertainty when lacking confidence in factual statements, significantly reducing hallucination rates through dampening prompts and factual calibration (Huang et al., 2025). Similarly, Ren et al. (2023) proposed using self-evaluation scores for selective generation, where LLMs decide when to abstain from generating, enhancing accuracy by reformulating open-ended tasks into token-level predictions.

DEEPCONF contributes to efficient reasoning by filtering low-confidence reasoning traces and enabling early termination based on local confidence signals, thereby optimizing computational overhead (Fu et al., 2024b).

Recent studies also examine the relationship between prediction mechanisms and training efficiency. Sieker et al. (2024) demonstrate that gains from complex Learning from AI Feedback (LAIF) pipelines may be overestimated, finding that Supervised Fine-Tuning (SFT) with a strong teacher model can be surprisingly effective. This implicitly supports the value of understanding evaluation outcomes to guide efficient training strategies.

**Our approach** in diverges significantly from these internal confidence mechanisms. Rather than focusing on self-assessment or selective generation, we aim to predict the scores assigned by *external* LLM or agentic judges to a model's response *before* generation occurs. This involves learning to anticipate evaluator judgments directly, encompassing both zero-shot and contextual predictions based on historical performance data. This pre-hoc prediction enables resource optimization without requiring real-time introspection or response generation.

For completeness, we discuss supplementary related work in Appendix A.

## 7 CONCLUSION

In this work, we examined the ability of language models of varying sizes to predict the quality of their own responses as judged by an external agentic (LLM-based) evaluator. Our results show that while larger models tend to be better calibrated in their self-assessments, smaller models often exhibit overconfidence or underconfidence. To address this, we proposed two approaches: providing models with report cards summarizing their historical performance, and fine-tuning models specifically for performance prediction. Both methods improved models' ability to estimate their own response quality, with the degree of improvement depending on model size and task difficulty Altogether, we believe that this work represents a step forward in increasing the usability of small language models.

## 8 LIMITATIONS & FUTURE WORK

Our evaluation is limited to a single judge model (Llama 3.3 70B and GPT OSS 120B) and single-turn interactions, which may not generalize to diverse evaluation frameworks or multi-turn conversations. Additionally, we use a relatively generic evaluation rubric with our judge. One of the key advantages of such judges is that they can integrate arbitrary alignment requirements into their evaluations. Future work will look at taking advantage of this as well as exploit more advanced judge models and architectures, and evaluate multi-turn interactions.

Another key limitation of this work is that the report card approach requires processing many tokens to generate a prediction. While models are typically faster at processing input tokens than generating output tokens, this still introduces an additional overhead. Likewise, the fine-tuning approach necessitates either maintaining a second set of weights or sacrificing model performance. Future work will look at integrating the predictions directly into the models so that the same prompt can be used for both prediction and response generation. In addition, future work will also explore alternative training strategies, such as reinforcement learning from judge feedback or leveraging human-in-the-loop corrections.

ETHICS STATEMENT

The authors see no outstanding ethical issues of particular concern that could arise from this paper.

REPRODUCIBILITY STATEMENT

All experiments were performed on one machine with eight (8) NVIDIA H100 chips. Replicating all the experiments run in the paper requires approximately forty-eight (48) hours on such a machine. All of the datasets and models were taken from HuggingFace directly. Existing splits in the datasets were merged, and then a 75/25 split was made (using a static seed) to produce a training and testing set (hence the ranges of values in the value of $n$ in several plots). The distribution of judge scores and zero-shot results in the paper refer to the training set. The contextual and fine-tuning results in the paper refer to the testing set. The full source code to replicate our experiments is available at REDACTED

AUTHOR CONTRIBUTIONS

REDACTED

ACKNOWLEDGMENTS

REDACTED

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

# A SUPPLEMENTARY RELATED WORKS

Beyond the core areas of LLM evaluation and performance prediction discussed in the main text, several adjacent research directions provide important context for our work.

**Prompt Engineering and Optimization.**   Substantial research focuses on optimizing LLM interactions through sophisticated prompt engineering techniques. Such prompt engineering strategies date back to the work on Learning to Think (Schmidhuber, 2015) (later refined (Schmidhuber, 2018)), wherein one network learns how to query and extract information from another network. Evolutionary algorithms offer a gradient-free approach for prompt optimization in black-box scenarios by evolving effective instructions and demonstrations (Guo et al., 2024; Fernando et al., 2024).

**Benchmarks for LLM and Agent Evaluation.**   The rapid advancement of LLMs and agentic systems has necessitated continuous development of comprehensive benchmarks. These span from basic code generation (HumanEval (Chen et al., 2021), MBPP (Austin et al., 2021)) to complex software engineering tasks (SWE-Bench (Jimenez et al., 2024), DevBench (Li et al., 2024a)). For machine learning workflows specifically, benchmarks like DevAI (Zhuge et al., 2024), ML-Dev-Bench (Padigela et al., 2025), and ML-BENCH evaluate end-to-end capabilities including environment setup and API integration, posing distinct challenges for current models (Tang et al., 2024). Recent multi-agent evaluation frameworks (Liu et al., 2024a) further extend these capabilities to collaborative settings.

**Retrieval-Augmented Generation (RAG) Systems.**   RAG systems are extensively developed to enhance LLMs' factual accuracy and contextual understanding, incorporating advanced strategies for chunking, metadata enrichment, and confidence-based retrieval triggering (Lewis et al., 2020; Huang et al., 2025). Recent advancements include adaptive retrieval mechanisms (Jeong et al., 2024) and multi-hop reasoning frameworks (Liu et al., 2025) that improve information integration.

**Model Efficiency and Optimization.**   Parallel research focuses on optimizing LLM inference through techniques such as speculative decoding (Leviathan et al., 2023) and quantization methods (Frantar et al., 2022). While these approaches target computational efficiency during generation, our work addresses efficiency at the evaluation level by predicting scores without requiring full response generation.

# B  SUMMARY TABLE FOR CONTEXTUAL RESULTS

Table 2 summarizes the improvement in prediction accuracy for the contextual setting as compared with the zero-shot setting. For a summary of the fine-tuned model results, see Figure 13.

| Model | MedQA | LongFact | AIME 2024 | MMLU-Pro | SciCode | *Mean* |
|---|---|---|---|---|---|---|
| M09B | -0.10 | +0.27 | -0.15 | -0.11 | -0.43 | -0.10 |
| L321B | +0.28 | +0.12 | +0.91 | +0.55 | +0.90 | **+0.55** |
| L323B | -0.01 | -0.12 | +0.35 | +0.24 | -0.12 | +0.07 |
| L318B | +0.22 | -0.05 | -0.22 | -0.00 | +0.03 | -0.00 |
| DSQ14B | +0.14 | +0.07 | -0.17 | +0.07 | -0.02 | +0.02 |
| GPT20B | +0.00 | -0.13 | +0.16 | -0.01 | -0.08 | -0.01 |
| DSQ32B | +0.03 | +0.09 | +0.75 | +0.02 | -0.13 | +0.15 |
| L3370B | +0.59 | +0.67 | -0.03 | +0.24 | +0.38 | +0.37 |
| DSL70B | +0.14 | +0.21 | +0.53 | +0.10 | -0.22 | +0.15 |
| L416E | +0.06 | +0.08 | +0.01 | +0.13 | +0.28 | +0.11 |
| GPT120B | +0.04 | +0.18 | +0.05 | -0.05 | +0.18 | +0.08 |

Table 2: Summary of the improvement in the prediction accuracy for the contextual setting as compared with the zero-shot setting.

## C   PROMPTS AND PROMPT TEMPLATES

Prompts 1 to 11 and 13 shows the prompts used for different parts of the system. The contents enclosed in curly brackets are substituted by the relevant content while the experiment is running.

```
You are a helpful AI assistant that answers user queries. You are given a user query
and you must answer the query in a helpful and concise manner (1-3 sentences).
Responses that are too long are considered to be very bad responses and will be
additionally truncated for the user. You must not ask any follow up questions and
should assume this is the only question that will be asked. You must not ask for
clarification, more information, more context, or more details.
```

Prompt 1: The system prompt used by the models when asked to respond to a query or estimate how the LLM judge will score their response.

```
You are an expert AI evaluator who grades the quality of responses to user queries.
Your job is to carefully analyze AI responses and grade each one. You will assign
each response one of these grades:
- great: Excellent response that is accurate, helpful, concise, and directly
addresses the query
- ok: Adequate response with minor issues in accuracy, helpfulness, or clarity
- bad: Poor response with significant issues in accuracy, helpfulness, or clarity,
or severely inadequate response with major errors or completely fails to address the
query

Follow the output format specified in the instructions exactly.
```

Prompt 2: The system prompt used by the LLM judge.

```
You are given a user query. Predict how well you would answer this query by
estimating what score a judge would give to an answer you would provide. Do this
without attempting to actually answer the query.

**IMPORTANT INSTRUCTIONS**
You must output EXACTLY ONE WORD and NOTHING ELSE. Do NOT attempt to actually answer
the query; ONLY output a score prediction.

**USER QUERY**
The user gave this query: {query}

**RESPONSE FORMAT**
Your entire response must be ONLY ONE of these three words:
- great
- ok
- bad

Do NOT include any other text, explanation, or answer to the query itself. Do NOT
add quotes, punctuation, or any additional characters to your response.
```

Prompt 3: The prompt used to ask a model to predict the LLM judge's score for its future response zero-shot.

```
You are given a user query and feedback from your previous performance on similar
queries. Use this feedback to predict how well you would answer this query by
estimating what score a judge would give to an answer you would provide. Do this
without attempting to actually answer the query.

**IMPORTANT INSTRUCTIONS**
You must output EXACTLY ONE WORD and NOTHING ELSE. Do NOT attempt to actually answer
the query; ONLY output a score prediction.

**PREVIOUS PERFORMANCE FEEDBACK**
Here is feedback from your performance on previous queries and datasets:
{model_feedback}

**USER QUERY**
The user gave this query: {query}

**ANALYSIS GUIDANCE**
Consider the following when making your prediction:
- Look for similar query types or domains in your feedback
- Identify patterns in your strengths and weaknesses from the feedback
- Consider how the current query relates to areas where you performed well or poorly
- Use the most relevant feedback to inform your prediction

**RESPONSE FORMAT**
Your entire response must be ONLY ONE of these three words:
- great
- ok
- bad

Do NOT include any other text, explanation, or answer to the query itself. Do NOT
add quotes, punctuation, or any additional characters to your response.
```

Prompt 4: The prompt used to ask a model to predict the LLM judge's score for its future response based on a report card for the model.

You were tested on the AIME 2024 dataset, which features high-level competition
mathematics problems from the American Invitational Mathematics Examination that
require sophisticated mathematical problem-solving skills and multi-step reasoning
to arrive at precise numerical answers. On this dataset, most of your responses were
judged to be "{HuggingFaceH4/aime_2024}". This means that your ability to solve
competition-level mathematics problems that require sophisticated problem-solving
skills and multi-step reasoning is "{HuggingFaceH4/aime_2024}".

You were tested on the LongFact dataset, which presents concept-based queries that
demand comprehensive, factual responses on topics like 20th-century events, US
foreign policy, accounting principles, and architecture, testing your ability to
provide detailed, accurate information across broad knowledge domains. On this
dataset, most of your responses were judged to be "{claserken/longfact}". This means
that your ability to regurgitate factual trivia is "{claserken/longfact}".

You were tested on the MedQA dataset, which focuses specifically on medical
question-answering with free-form multiple-choice questions derived from
professional medical board exams, testing clinical knowledge and diagnostic
reasoning. On this dataset, most of your responses were judged to be
"{bigbio/med_qa}". This means that your ability to diagnose medical conditions is
"{bigbio/med_qa}".

You were tested on the MMLU-Pro dataset, which contains challenging
undergraduate-level multiple-choice questions across diverse academic disciplines
including mathematics, science, history, and humanities, designed to test advanced
reasoning and knowledge beyond standard benchmarks. On this dataset, most of your
responses were judged to be "{TIGER-Lab/MMLU-Pro}". This means that your ability to
answer general undergraduate-level examination questions is "{TIGER-Lab/MMLU-Pro}".
You were further scored on different academic subjects within the MMLU-Pro dataset,
with the following results:
- Philosophy: {TIGER-Lab/MMLU-Pro/philosophy}
- Mathematics: {TIGER-Lab/MMLU-Pro/math}
- Economics: {TIGER-Lab/MMLU-Pro/economics}
- Engineering: {TIGER-Lab/MMLU-Pro/engineering}
- Physics: {TIGER-Lab/MMLU-Pro/physics}
- Biology: {TIGER-Lab/MMLU-Pro/biology}
- Business: {TIGER-Lab/MMLU-Pro/business}
- History: {TIGER-Lab/MMLU-Pro/history}
- Law: {TIGER-Lab/MMLU-Pro/law}
- Health: {TIGER-Lab/MMLU-Pro/health}
- Chemistry: {TIGER-Lab/MMLU-Pro/chemistry}
- Computer Science: {TIGER-Lab/MMLU-Pro/computer_science}
- Other subjects: {TIGER-Lab/MMLU-Pro/other}
These reflect how your ability to answer undergraduate-level examination questions
is affected by the subject.

You were tested on the SciCode dataset, which evaluates scientific computing and
programming capabilities through complex problem-solving tasks that require
understanding of scientific concepts, mathematical reasoning, and code
implementation across various scientific domains. On this dataset, most of your
responses were judged to be "{SciCode1/SciCode}". This means that your ability to
write source code for scientific work is "{SciCode1/SciCode}".

Prompt 5: The sub-prompt used to encode the report card for a model.

```
Your ability to solve competition-level mathematics problems that required
sophisticated problem-solving skills and multi-step reasoning is
"{HuggingFaceH4/aime_2024}". Your ability to regurgitate factual trivia is
"{claserken/longfact}". Your ability to diagnose medical conditions is
"{bigbio/med_qa}". Your ability to answer general undergraduate-level examination
questions is "{TIGER-Lab/MMLU-Pro}". Your ability to write source code for
scientific work is "{SciCode1/SciCode}".
```

Prompt 6: A short variant of the sub-prompt in Prompt 5 used to encode the report card for a model. See Appendix H for the ablation results using this prompt.

```
You are given a user query and feedback about an arbitrary model's previous
performance on similar queries. Use this feedback to predict how well that model
would answer this query by estimating what score a judge would give to an answer the
model would provide. Do this without attempting to actually answer the query.

**IMPORTANT INSTRUCTIONS**
You must output EXACTLY ONE WORD and NOTHING ELSE. Do NOT attempt to actually answer
the query; ONLY output a score prediction.

**MODEL PERFORMANCE FEEDBACK**
Here is feedback from the model's performance on previous queries and datasets:
{model_feedback}

**USER QUERY**
The user gave this query: {query}

**ANALYSIS GUIDANCE**
Consider the following when making your prediction:
- Look for similar query types or domains in the model's feedback
- Identify patterns in the model's strengths and weaknesses from the feedback
- Consider how the current query relates to areas where the model performed well or
poorly
- Use the most relevant feedback to inform your prediction about the model's likely
performance

**RESPONSE FORMAT**
Your entire response must be ONLY ONE of these three words:
- great
- ok
- bad

Do NOT include any other text, explanation, or answer to the query itself. Do NOT
add quotes, punctuation, or any additional characters to your response.
```

Prompt 7: The prompt used to ask a model to predict the LLM judge's score for an arbitrary model's future response based on a report card for the model.

```
Your prediction must be exactly one of these three values: 'great', 'ok', or 'bad'.
Please provide only one of these values.
```

Prompt 8: The prompt used to request a model to correct its prediction if the parser was unable to read it.

```
I'll provide you with a user query and {num_responses} AI responses to that query.
Please evaluate each response and grade each of them as either great, ok, or bad.

**RESPONDER INSTRUCTIONS**
The AI responders were given these specific instructions to follow when responding
to the user query:
{responder_system_prompt}

**EVALUATION RUBRIC**
Here is the rubric you will use to evaluate the AI responses:
{rubric}

You must strictly adhere to the rubric above when evaluating. Base your grades
entirely on the criteria specified in the rubric.

**USER QUERY**
The user gave this query: {query}

**HINTS**
The dataset says the following about how to answer the query:
{hints}

**AI RESPONSE**
The AIs responded with these responses:
{responses}

**RESPONSE FORMAT**
You should write an explanation of your evaluation for each model according to each
of the criteria in the rubric. If instruction following is in the rubric, then you
should look at each of the instructions given and say whether or not they were
followed. If the rubric asks about factual knowledge, then you should look at each
given fact and say whether or not they were correct. You should then explain what
final grade the rubric would assign to the output and why. When you're done giving
your justifications, output the token </rationale> and then the final grades for
each model---and only the final grades for the models---according to the following
JSON format:
{json_format}

Use only these grades in your evaluation: great, ok, bad. Output valid JSON for the
final grades only (don't forget the curly braces). Don't forget to output the
</rationale> token before the final grades.
```

Prompt 9: The prompt used when asking the LLM judge to simultaneously grade the responses of all the models to a single query.

```
I'll provide you with a user query and an AI response to that query. Please evaluate
the response and grade it as either great, ok, or bad.

**RESPONDER INSTRUCTIONS**
The AI responder was given these specific instructions to follow when responding to
the user query:
{responder_system_prompt}

**EVALUATION RUBRIC**
Here is the rubric you will use to evaluate the AI response:
{rubric}

You must strictly adhere to the rubric above when evaluating. Base your grade
entirely on the criteria specified in the rubric.

**USER QUERY**
The user gave this query: {query}

**HINTS**
The dataset says the following about how to answer the query:
{hints}

**AI RESPONSE**
The AI responded with this response: {response}

**RESPONSE FORMAT**
You should write an explanation of your evaluation for the model according to each
of the criteria in the rubric. If instruction following is in the rubric, then you
should look at each of the instructions given and say whether or not they were
followed. If the rubric asks about factual knowledge, then you should look at each
given fact and say whether or not they were correct. You should then explain what
final grade the rubric would assign to the output and why. When you're done giving
your justifications, output the token </rationale> and then the final grade for the
model---and only the final grade for the model---according to the following JSON
format:
{json_format}

Use only these grades in your evaluation: great, ok, bad. Output valid JSON for the
final grade only (don't forget the curly braces). Don't forget to output the
</rationale> token before the final grade.
```

Prompt 10: A variant of the prompt in Prompt 9 used when asking the LLM judge to grade the responses of a single model to a single query. See Appendix E for the ablation results using this prompt.

```
| Criteria | Great | Ok | Bad |
|----------|-------|----|----|
| **Accuracy** | Information is completely correct with no factual errors |
Information is mostly correct with minor inaccuracies that don't affect the main
point | Information contains significant factual errors or misleading statements |
| **Relevance** | Directly addresses the query with focused, on-topic content |
Addresses the query but includes some tangential or less relevant information |
Fails to address the main query or provides largely irrelevant content |
| **Completeness** | Fully answers all aspects of the question with appropriate
depth | Answers most aspects of the question but may miss minor details | Provides
incomplete answer, missing key information or major aspects |
| **Clarity** | Clear, well-structured, easy to understand with logical flow, and
ends gracefully | Generally clear but may have minor issues with structure or
explanation | Confusing, poorly structured, difficult to understand, or ends
abruptly |
| **Instruction Following** | Perfectly follows all given instructions and
constraints | Follows most instructions with minor deviations that don't impact
quality | Fails to follow key instructions or ignores important constraints |
| **Formatting** | Answer is perfectly formatted as requested with no extraneous
content | Answer is mostly formatted as requested but contains no extraneous content
| Answer is incorrectly formatted or contains any extraneous content not requested |

**Grading Guidelines:**
- **Great**: Meets ALL criteria at the "Great" level
- **Ok**: Meets most criteria at "Great" or "Ok" level, with no "Bad" ratings
- **Bad**: Has one or more criteria rated as "Bad" OR fails to meet minimum
standards across multiple criteria
```

Prompt 11: The rubric used by the LLM judge to grade responses.

```
| Criteria | Great | Ok | Bad |
|----------|-------|----|----|
| **Inaccuracy** | Information contains significant factual errors or misleading
statements | Information is mostly correct with minor inaccuracies that don't affect
the main point | Information is completely correct with no factual errors |
| **Irrelevance** | Fails to address the main query or provides largely irrelevant
content | Addresses the query but includes some tangential or less relevant
information | Directly addresses the query with focused, on-topic content |
| **Incompleteness** | Provides incomplete answer, missing key information or major
aspects | Answers most aspects of the question but may miss minor details | Fully
answers all aspects of the question with appropriate depth |
| **Incoherence** | Confusing, poorly structured, difficult to understand, or ends
abruptly | Generally clear but may have minor issues with structure or explanation |
Clear, well-structured, easy to understand with logical flow, and ends gracefully |
| **Non-compliance** | Fails to follow key instructions or ignores important
constraints | Follows most instructions with minor deviations that don't impact
quality | Perfectly follows all given instructions and constraints |
| **Malformatting** | Answer is incorrectly formatted or contains any extraneous
content not requested | Answer is mostly formatted as requested but contains no
extraneous content | Answer is perfectly formatted as requested with no extraneous
content |

**Grading Guidelines:**
- **Great**: Meets ALL criteria at the "Great" level
- **Ok**: Meets most criteria at "Great" or "Ok" level, with no "Bad" ratings
- **Bad**: Has one or more criteria rated as "Bad" OR fails to meet minimum
standards across multiple criteria
```

Prompt 12: The mischievous rubric used by the LLM judge to grade responses for Section I.

```
Your output is invalid. You gave the following answer:
{previous_response}

The instructions given to you were:
{instructions}
```

Prompt 13: The prompt used to request the LLM judge to correct its evaluation if the parser was unable to read it.

# D MODE SCORES

Table 3 shows the mode scores for each of the models on each of the datasets. Likewise, Table 4 shows the mode scores for each of the models on each of the categories in the MMLU-Pro dataset. As would be expected from looking at Figures 3 to 7, a good variation is observed between the different models and datasets, implying that the results presented here generalize well.

| Model | AIME 2024 | LongFact | MedQA | SciCode | MMLU-Pro |
|---|---|---|---|---|---|
| M09B | bad | bad | bad | bad | bad |
| L321B | bad | ok | bad | bad | bad |
| L323B | bad | great | great | bad | bad |
| L318B | bad | ok | great | bad | great |
| GPT20B | bad | great | great | great | great |
| DSQ14B | great | great | ok | great | great |
| DSQ32B | bad | great | great | ok | great |
| GPT120B | bad | great | great | great | great |
| DSL70B | bad | great | great | ok | great |
| L3370B | great | great | great | ok | great |
| L416E | great | great | great | great | great |

Table 3: The mode scores for each model on each of the datasets.

| Model | Biology | Business | Chemistry | Computer Science | Economics | Engineering |
|---|---|---|---|---|---|---|
| M09B | bad | bad | bad | bad | bad | bad |
| L321B | bad | bad | bad | bad | bad | bad |
| L323B | great | bad | bad | bad | bad | bad |
| L318B | great | bad | bad | bad | great | bad |
| GPT20B | great | great | great | great | great | bad |
| DSQ14B | great | great | great | great | great | great |
| DSQ32B | great | great | bad | great | great | bad |
| GPT120B | great | great | great | great | great | great |
| DSL70B | great | great | great | great | great | great |
| L3370B | great | great | great | great | great | ok |
| L416E | great | great | great | great | great | great |

| Graph | Health | History | Law | Math | Other | Philosophy | Physics | Psychology |
|---|---|---|---|---|---|---|---|---|
| M09B | bad | bad | bad | bad | bad | bad | bad | bad |
| L321B | bad | bad | bad | bad | bad | bad | bad | bad |
| L323B | great | great | bad | bad | great | bad | bad | great |
| L318B | great | great | great | bad | great | great | bad | great |
| GPT20B | great | great | bad | great | great | great | great | great |
| DSQ14B | great | great | ok | great | great | great | great | great |
| DSQ32B | great | great | great | great | great | great | great | great |
| GPT120B | great | great | great | great | great | great | great | great |
| DSL70B | great | great | great | great | great | great | great | great |
| L3370B | great | great | great | great | great | great | great | great |
| L416E | great | great | great | great | great | great | great | great |

Table 4: The mode scores for each model on each of the categories in MMLU-Pro.

# E  INDEPENDENT EVALUATION RESULTS

LLMs are known to be overconfident (Huang et al., 2025; Ren et al., 2023). Thus, a relative assessment of model outputs is likely to have a higher variation than an independent assessment. To verify this, we conducted an ablation in which the model outputs are assessed separately. The results of this are shown in Figures 16 to 20. Intuitively, the combination of this and a mixture of small and large models renders a relative assessment more meaningful here.

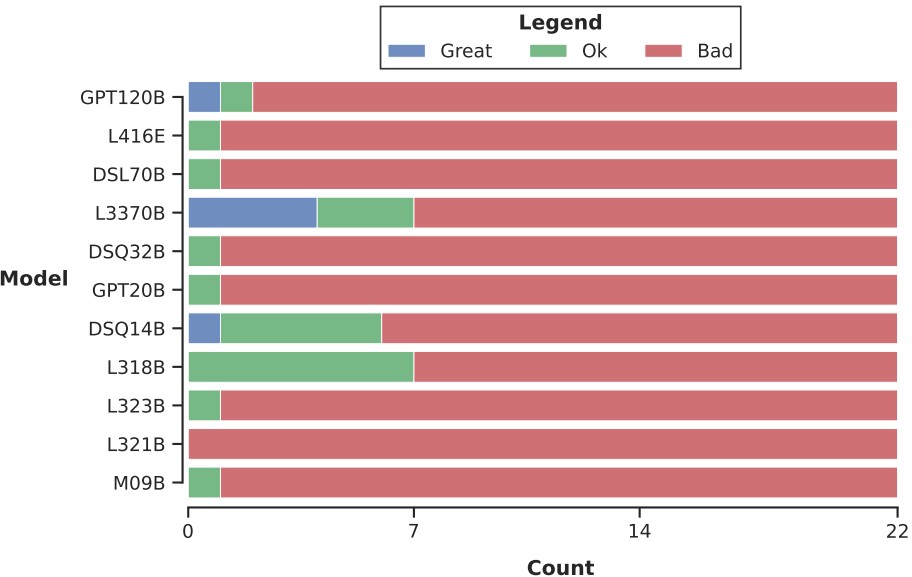

Figure 16: Distribution of LLM judge scores on AIME 2024 dataset under independent evaluation.

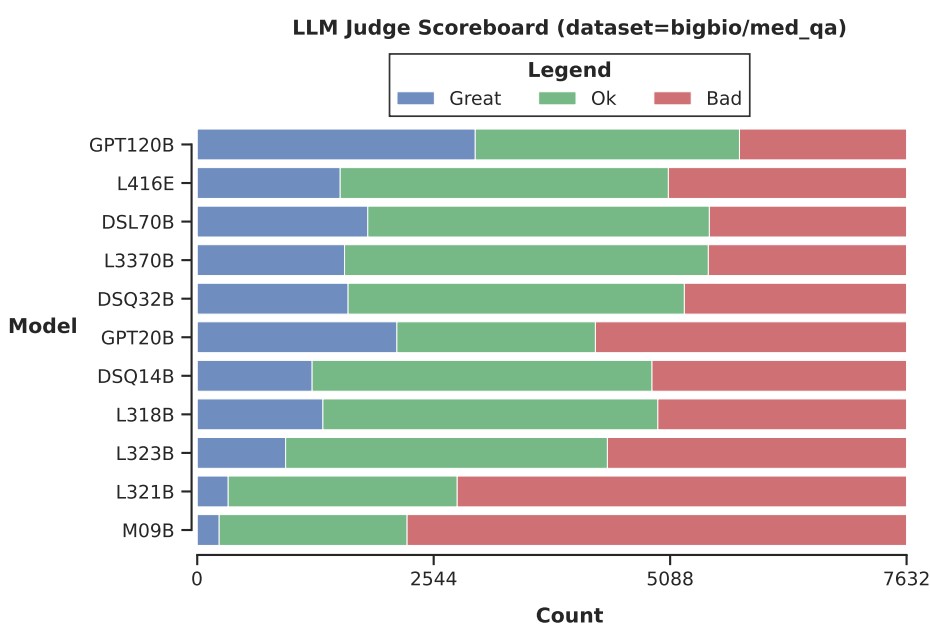

Figure 17: Distribution of LLM judge scores on MedQA dataset under independent evaluation.

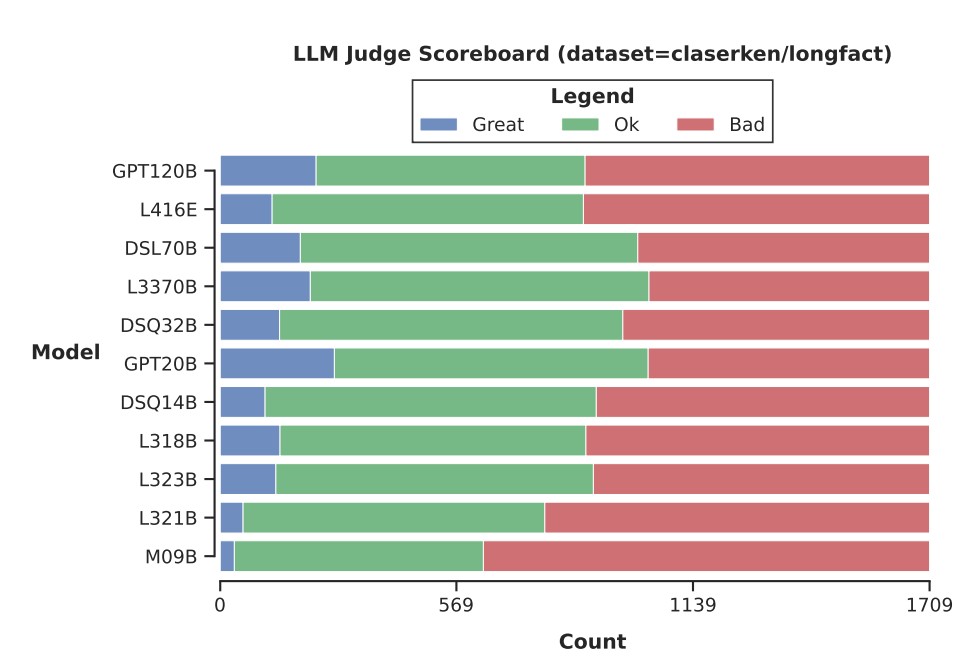

Figure 18: Distribution of LLM judge scores on LongFact dataset under independent evaluation.

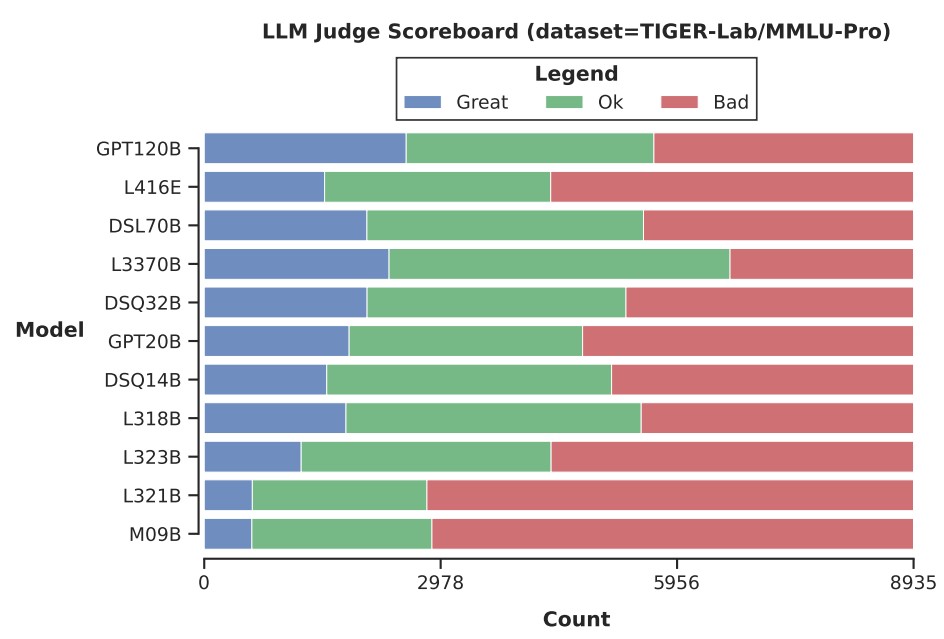

Figure 19: Distribution of LLM judge scores on MMLU-Pro dataset under independent evaluation.

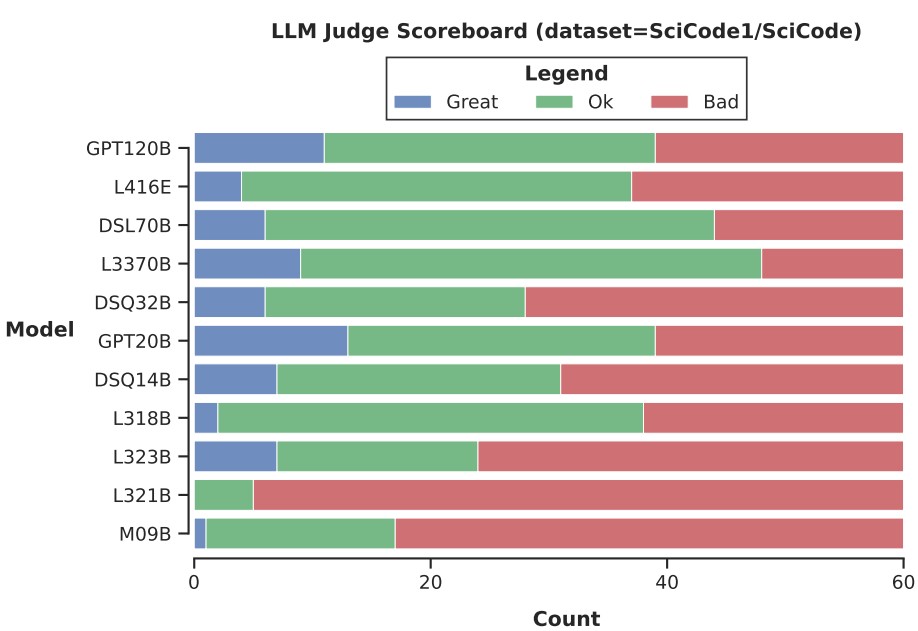

Figure 20: Distribution of LLM judge scores on SciCode dataset under independent evaluation.

## F    PREDICTION DISTRIBUTIONS

Figures 21 to 25 show the zero-shot and contextual prediction distributions for each of the models on each of the datasets. As expected, the distributions tend to have lower variance than the actual score distribution (as queries are being categorized coarsely).

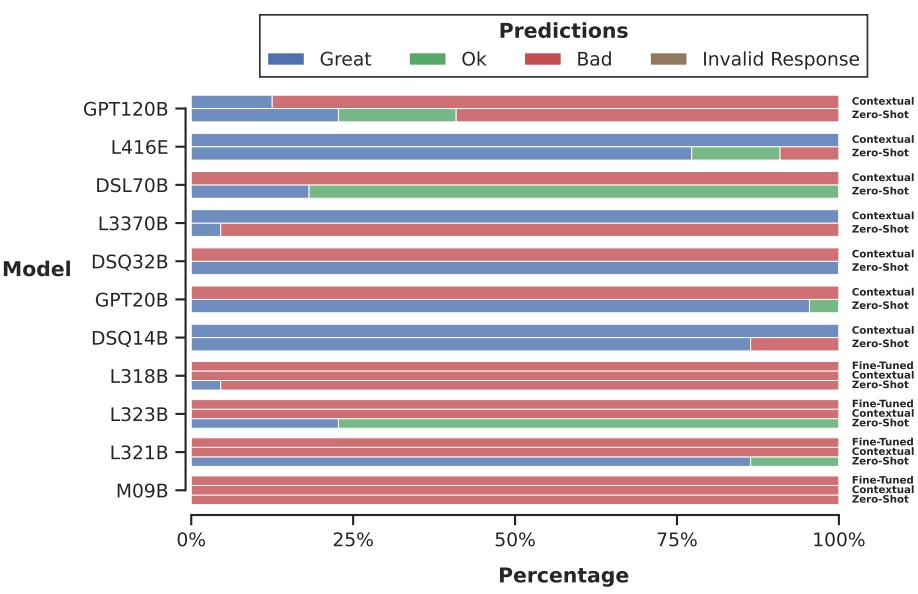

Figure 21: Zero-Shot and Contextual Prediction Distributions on AIME 2024

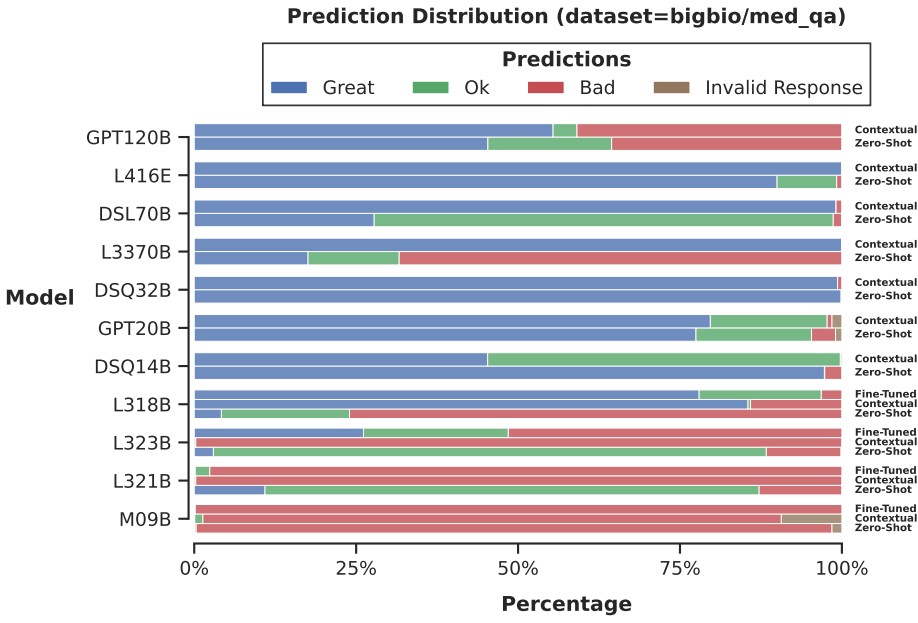

Figure 22: Zero-Shot and Contextual Prediction Distributions on MedQA

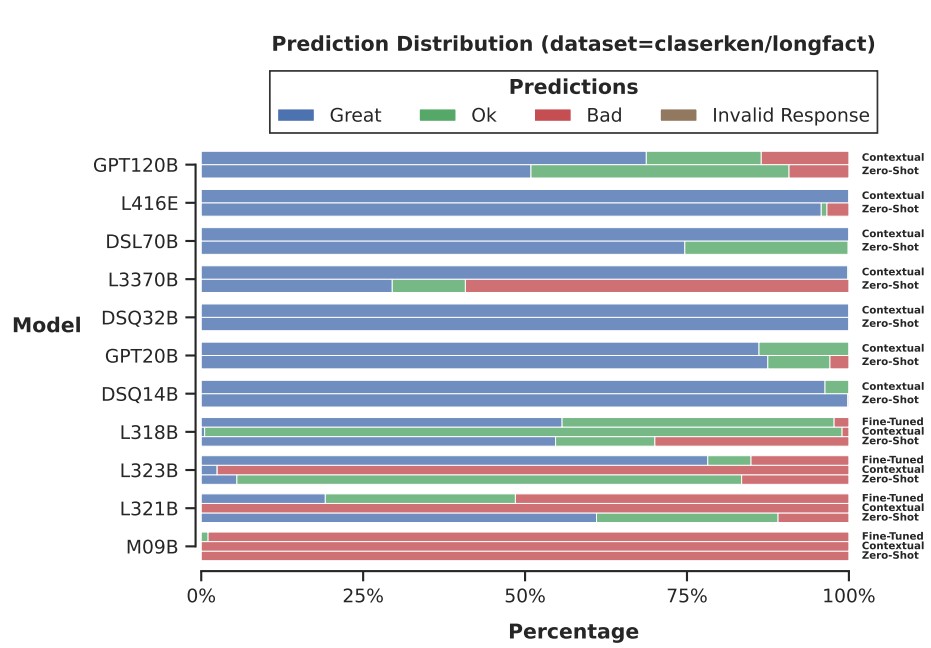

Figure 23: Zero-Shot and Contextual Prediction Distributions on LongFact

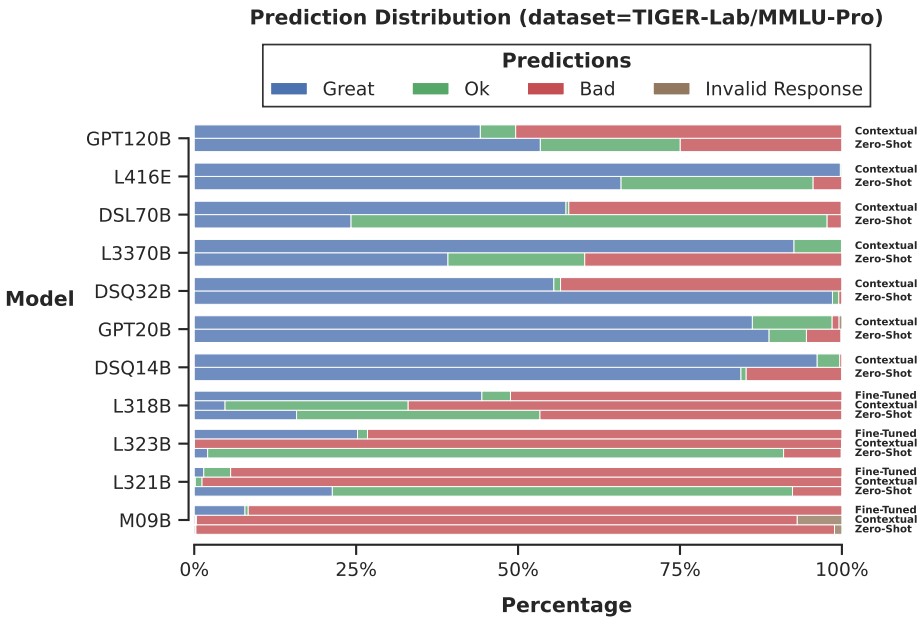

Figure 24: Zero-Shot and Contextual Prediction Distributions on MMLU-Pro

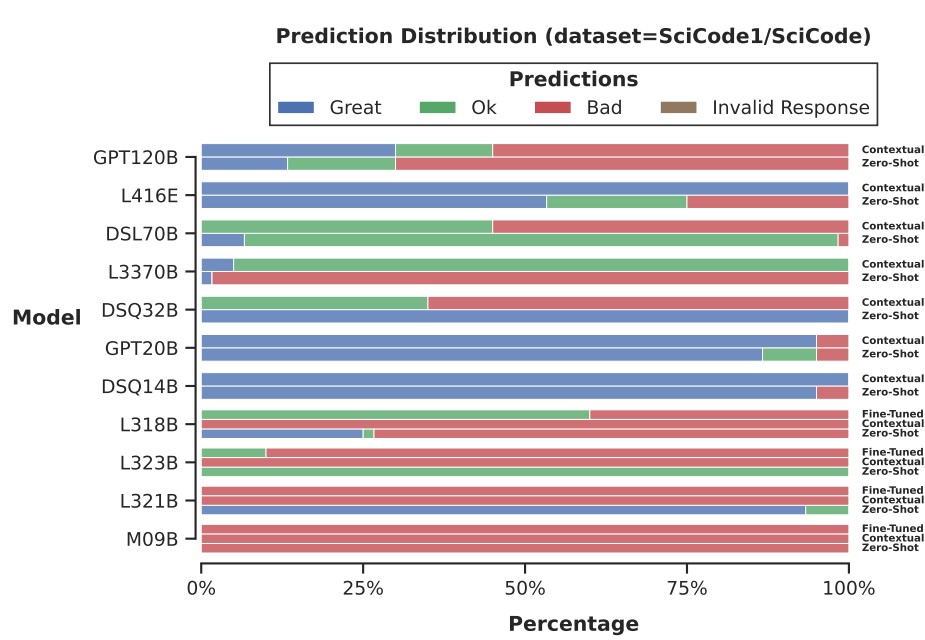

Figure 25: Zero-Shot and Contextual Prediction Distributions on SciCode

# G ALTERNATE JUDGE RESULTS

The use of Llama 3.3 70B as a judge was done as it's a well-studied and reliable large model. As such, the evaluations it gives to the output of different models can be safely assumed to be meaningful. This is sufficient for the goals of this work. However, to validate that this continues to hold if the judge model is trained, we repeated the experiments with GPT OSS 120B as the judge model, replacing it as a model to be evaluated by MobileLLM-R1 (Zhao et al., 2025) in order to keep the number of models being evaluated constant. The results are shown in Figures 1 through 10. While these results bear some similarities to the results in the main text, several differences are present. Most notably, GPT OSS 120B is a harsher judge than Llama 3.3 70B. Despite this, the report card strategy is still able to generally improve the predictive performance of the system.

We noted that when running these experiments, our experiences suggest that GPT OSS 120B is much less suitable as a judge, being prone to (1) low-variety blanket evaluations of responses, and (2) producing corrupted outputs where arbitrary thinking tokens not included in the specification for GPT OSS 120B are present. These observations are consistent with the fact that this task is most certainly outside the training regime of these models, and the trend of the newest generation of models (from around the time of Llama 4 onwards) is that they are heavily overfitted to benchmarks. A full analysis of this behaviour is outside the scope of this work.

Figure 26: Distribution of LLM judge scores on AIME 2024 dataset with GPT OSS 120B as the judge.

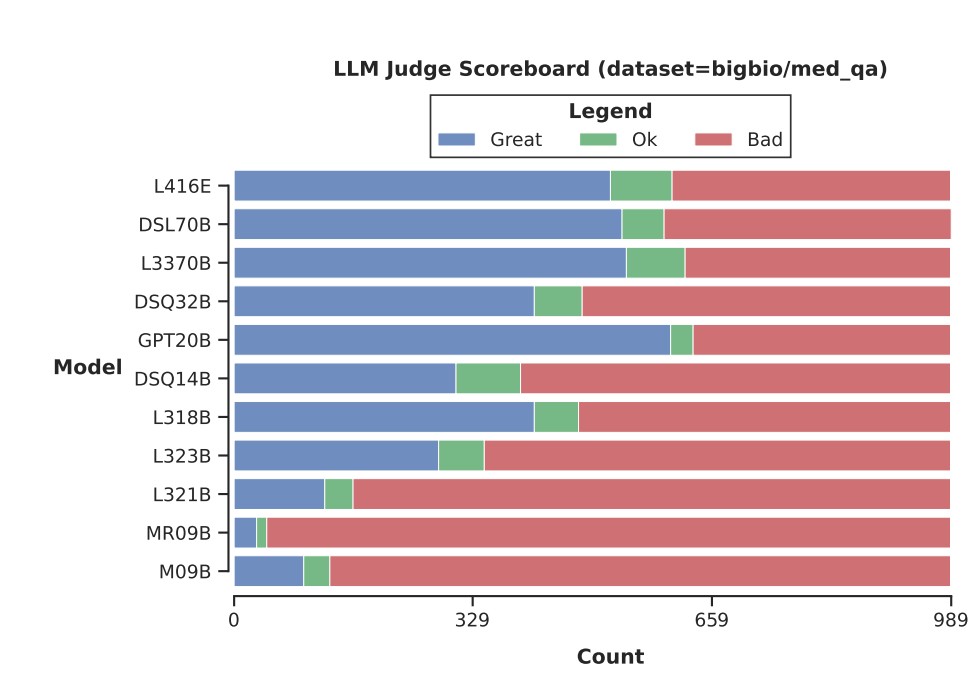

Figure 27: Distribution of LLM judge scores on MedQA dataset with GPT OSS 120B as the judge.

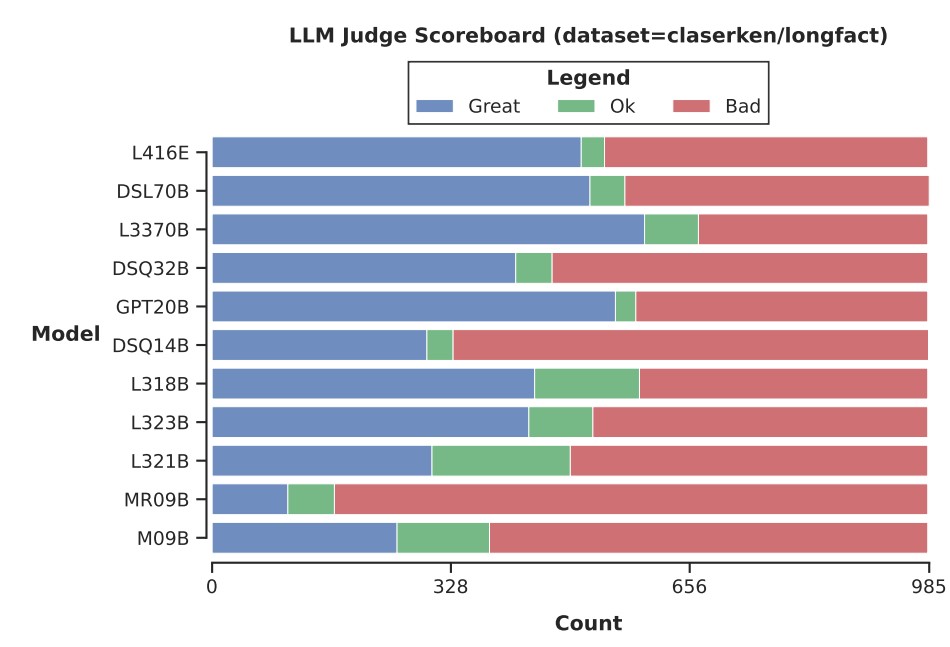

Figure 28: Distribution of LLM judge scores on LongFact dataset with GPT OSS 120B as the judge.

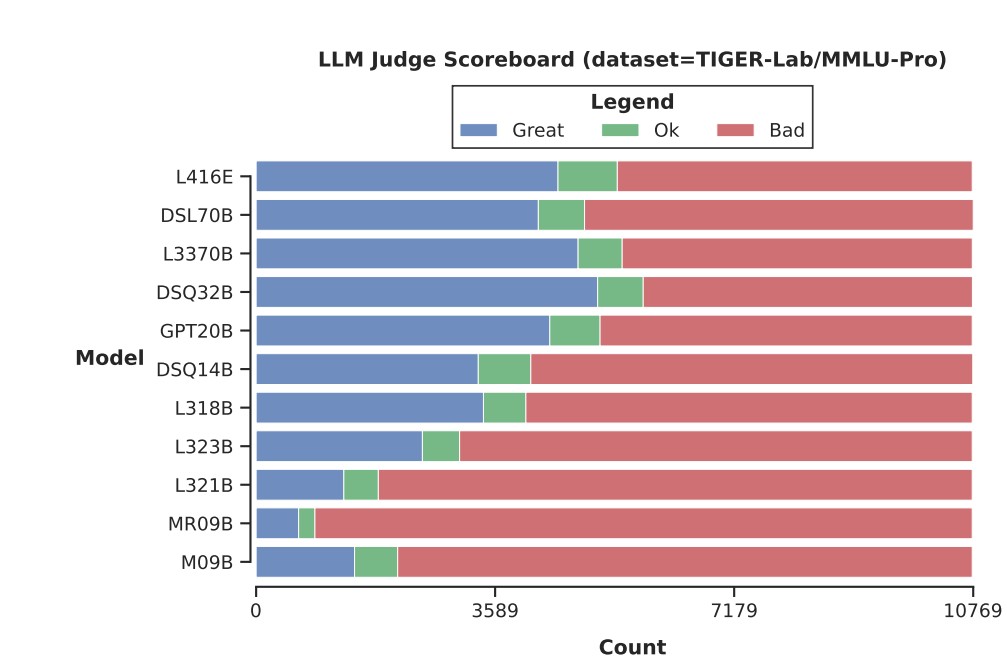

Figure 29: Distribution of LLM judge scores on MMLU-Pro dataset with GPT OSS 120B as the judge.

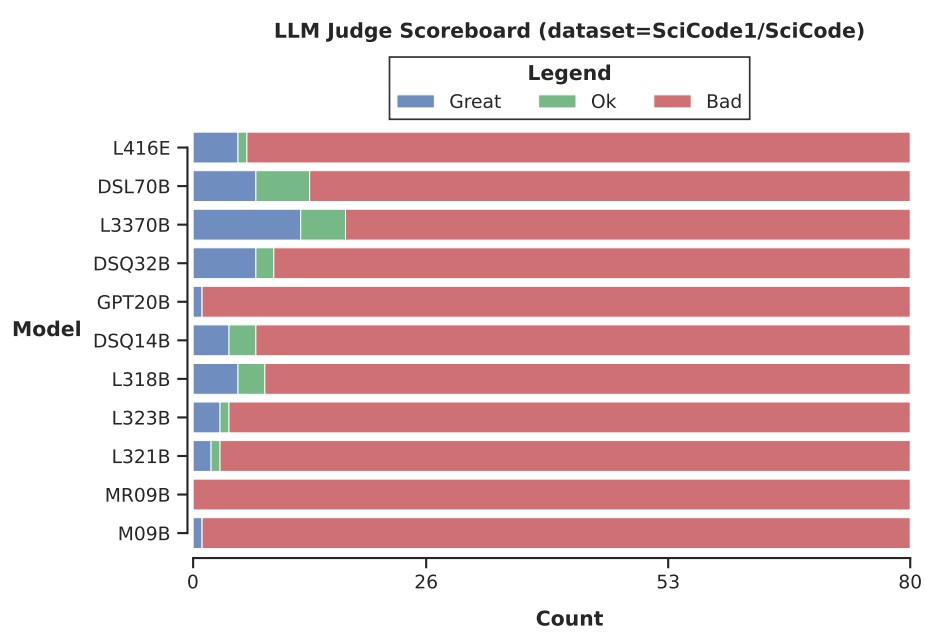

Figure 30: Distribution of LLM judge scores on SciCode dataset with GPT OSS 120B as the judge.

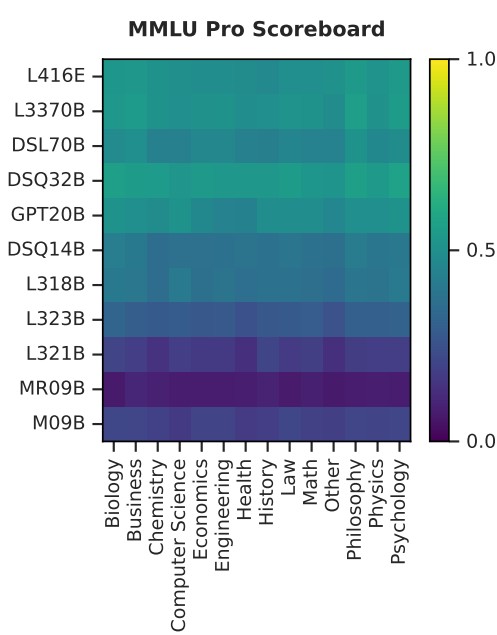

Figure 31: The probability of a model receiving a grade of either "great" or "ok" as a function of the query's category on the MMLU-Pro dataset when using GPT OSS 120B as the judge.

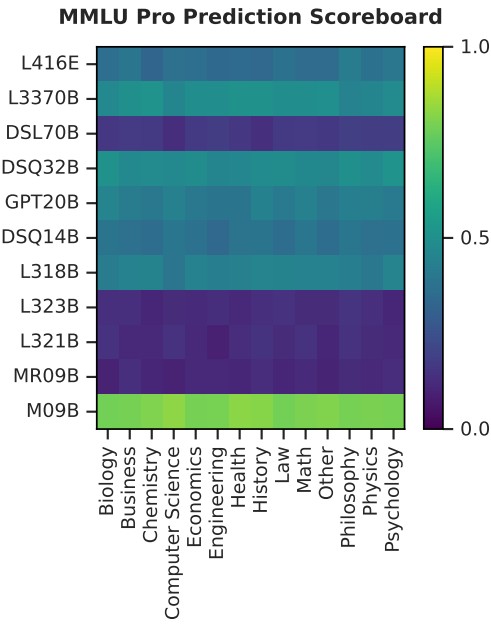

Figure 32: The probability of a model receiving a grade of either "great" or "ok" as a function of the query's category on the MMLU-Pro dataset when using GPT OSS 120B as the judge.

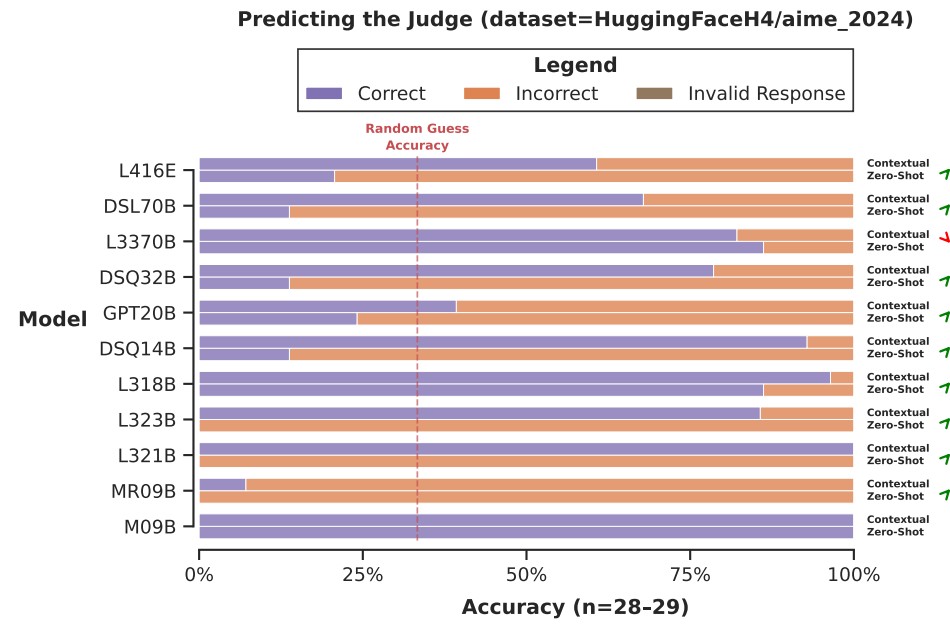

Figure 33: Contextual prediction accuracy of models on AIME 2024 dataset when using GPT OSS 120B as the judge.

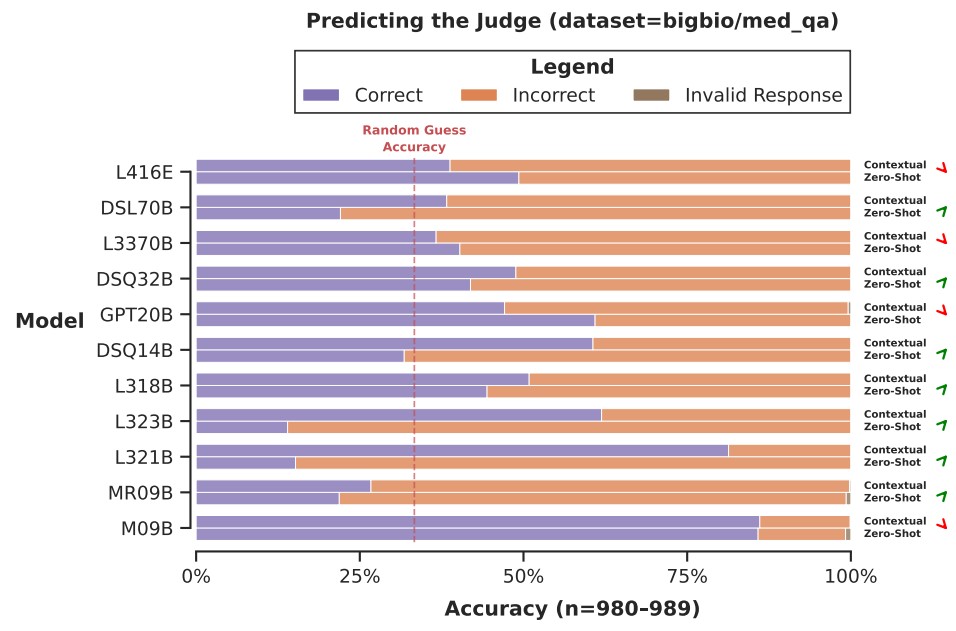

Figure 34: Contextual prediction accuracy of models on MedQA dataset when using GPT OSS 120B as the judge.

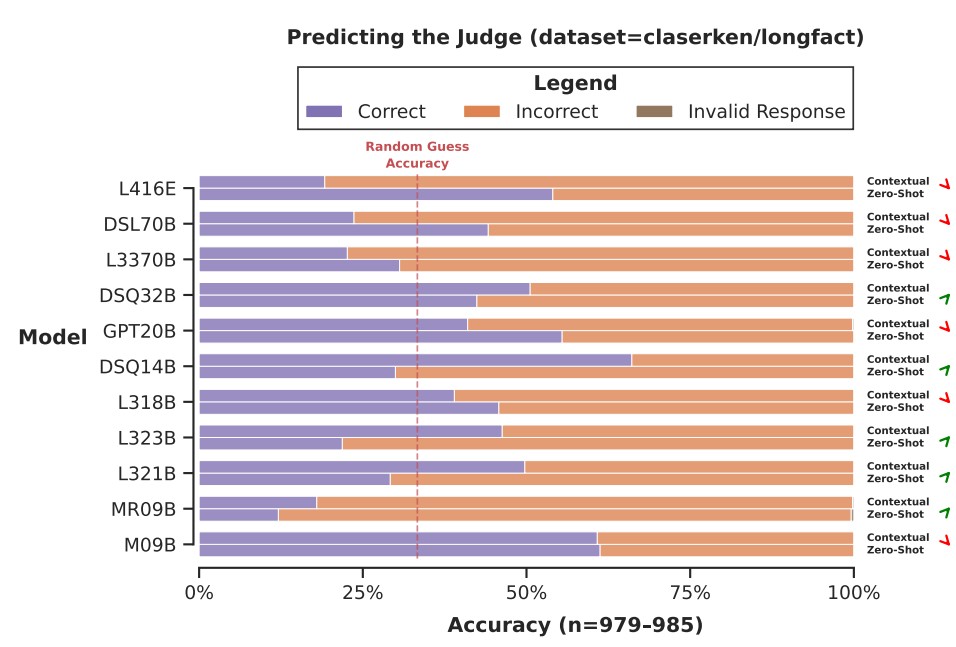

Figure 35: Contextual prediction accuracy of models on LongFact dataset when using GPT OSS 120B as the judge.

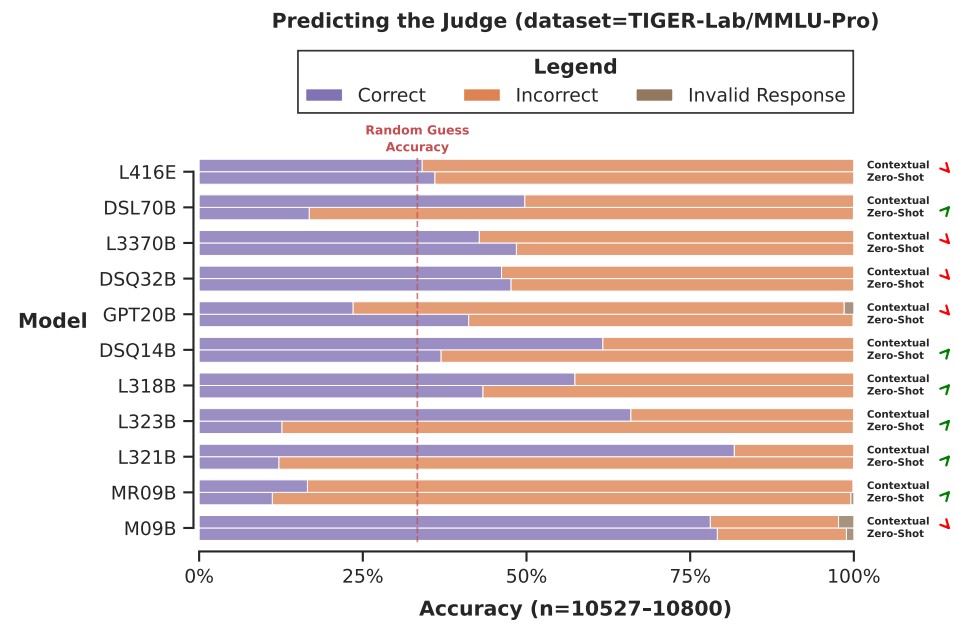

Figure 36: Contextual prediction accuracy of models on MMLU-Pro dataset when using GPT OSS 120B as the judge.

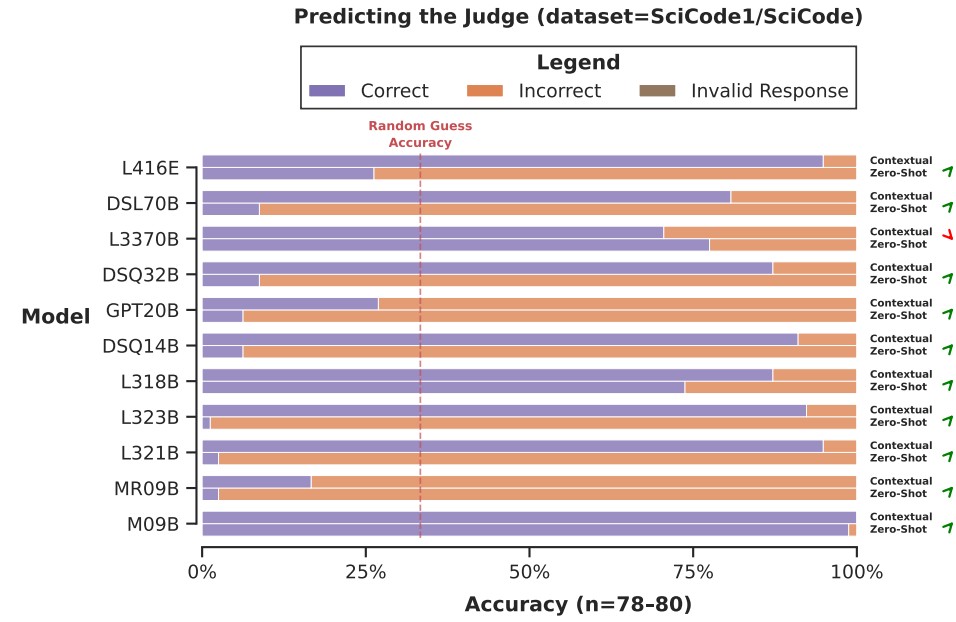

Figure 37: Contextual prediction accuracy of models on SciCode dataset when using GPT OSS 120B as the judge.

## H   SHORT FEEDBACK TEMPLATE RESULTS

The feedback template shown in Prompt 5 and used in our primary experiments is exceedingly long. To determine whether this is important or not, we repeated our zero-shot experiments with the shorter feedback template shown in Prompt 6. The results are shown in Figures 38 to 42. The principal takeaway from this ablation is that the longer and more comprehensive template led to a much stronger performance than the short template.

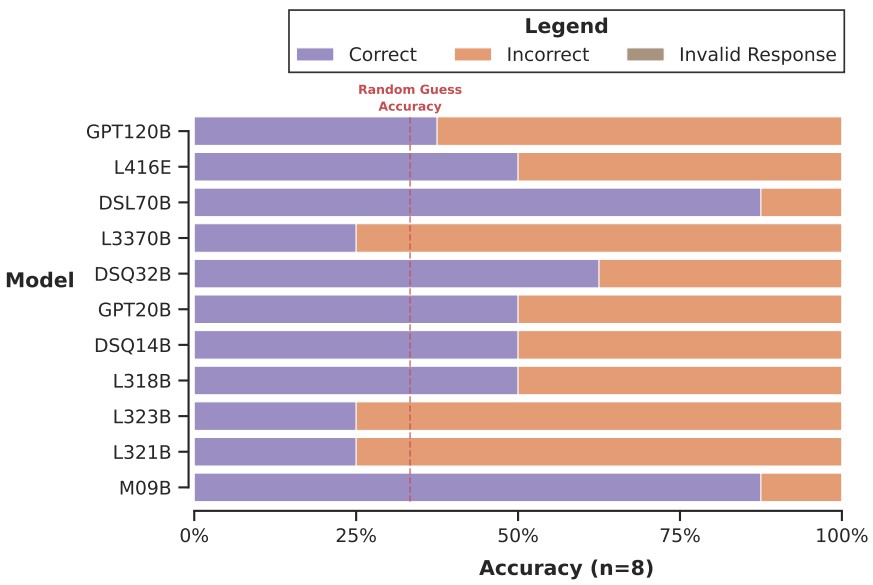

Figure 38: Contextual prediction accuracy of models on AIME 2024 dataset when using the short feedback template given in Prompt 6.

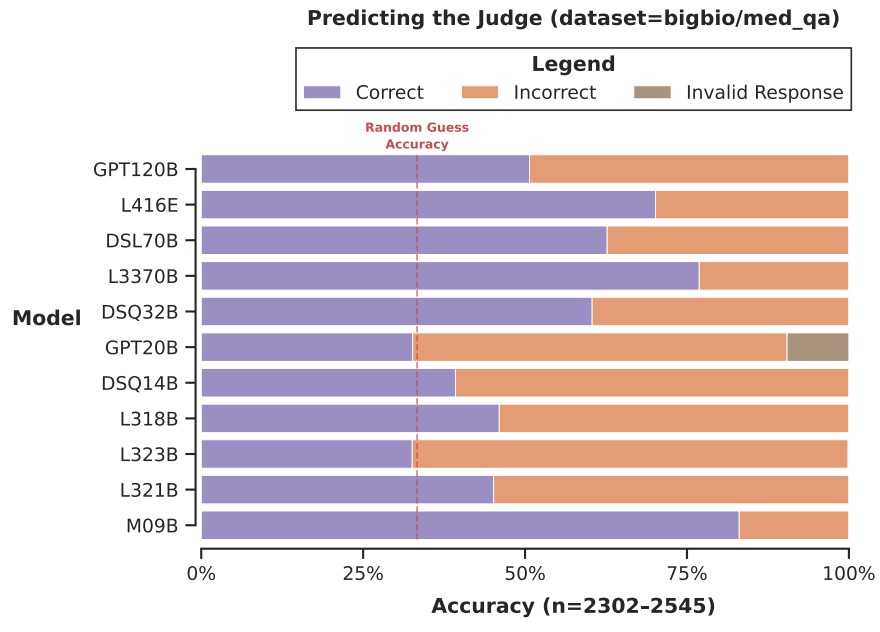

Figure 39: Contextual prediction accuracy of models on MedQA dataset when using the short feedback template given in Prompt 6.

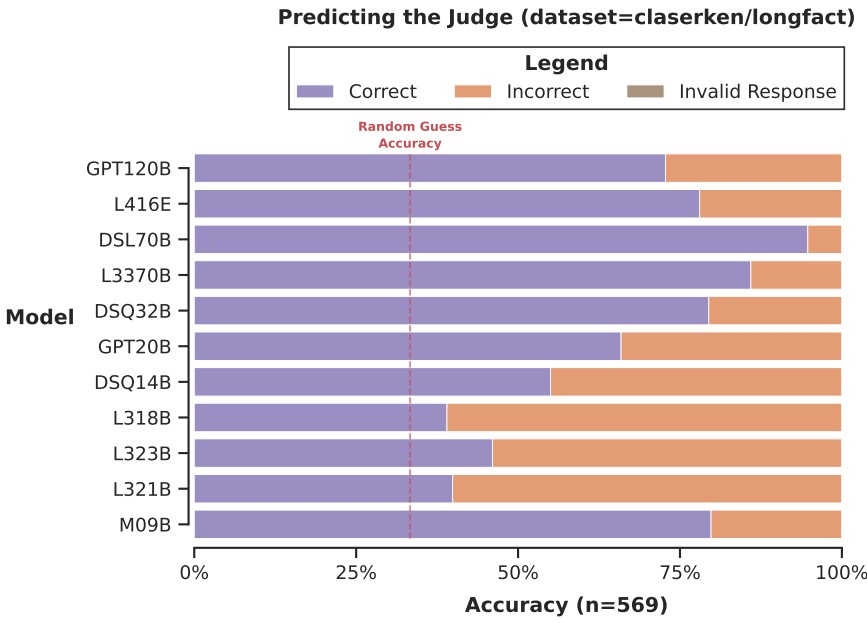

Figure 40: Contextual prediction accuracy of models on LongFact dataset when using the short feedback template given in Prompt 6.

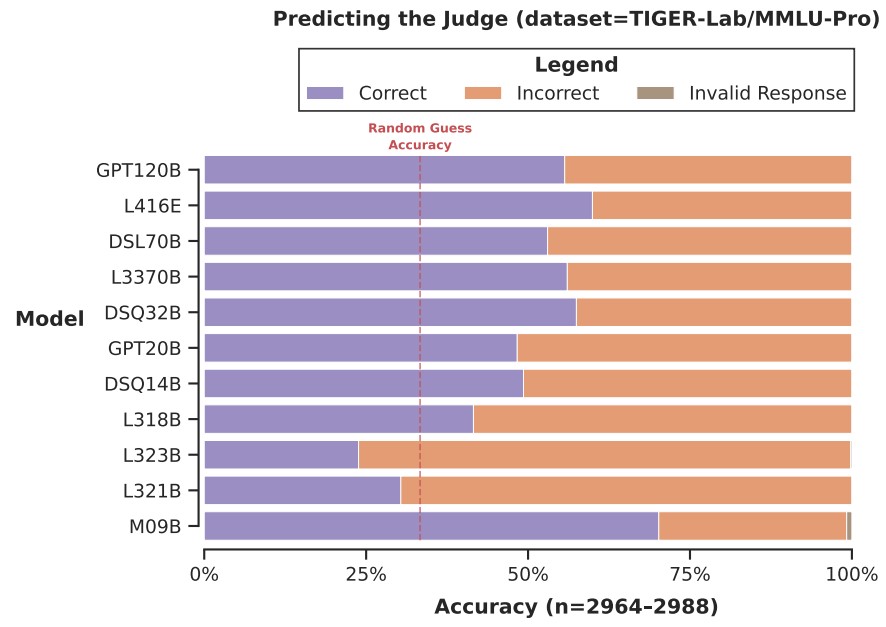

Figure 41: Contextual prediction accuracy of models on MMLU-Pro dataset when using the short feedback template given in Prompt 6.

Figure 42: Contextual prediction accuracy of models on SciCode dataset when using the short feedback template given in Prompt 6.

## I MISCHIEVOUS RUBRIC RESULTS

The rubric used in our primary experiments (and shown in Prompt 11) scores models according to a naive interpretation of performance. Part of the benefit of using a LLM or agentic judge is that you can define the evaluation criteria in a freeform manner. To demonstrate that this flexibility carries over to the prediction task, we repeated our experiment using an alternative, reversed rubric for evaluating the models. This rubric is shown in Prompt 12 with the rest of the prompt template used in Prompt 7. The results of this experiment are shown in Figures 43 to 47. While the predictive ability of the models given in Figures 43 to 47 is clearly less than than those given in Figures 8 to 12, a clear (better than random) predictive ability is displayed by most of the models on most of the datasets. This suggests that—in addition to using the report cards—the models are relying on their intrinsic understanding of themselves or the difficulty of the questions.

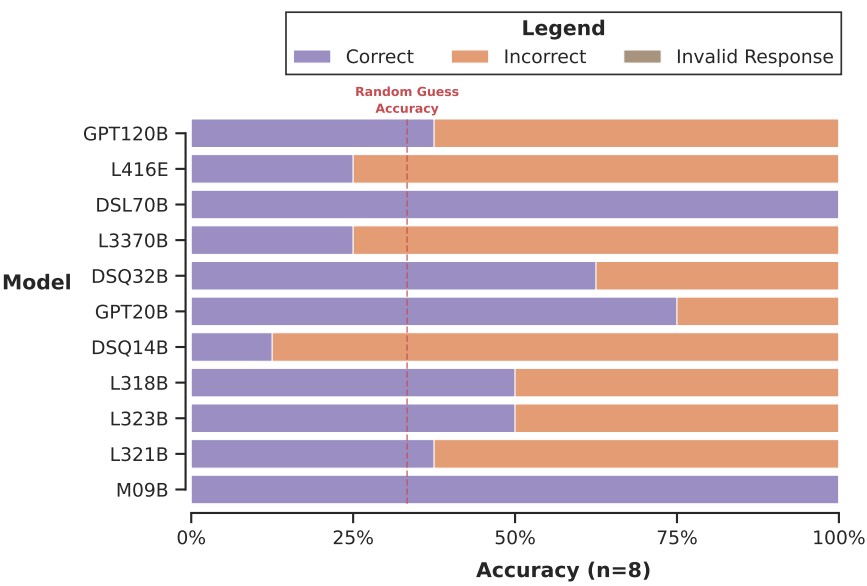

Figure 43: Contextual prediction accuracy of models on AIME 2024 dataset when evaluation is performed according to the mischievous rubric given in Prompt 12.

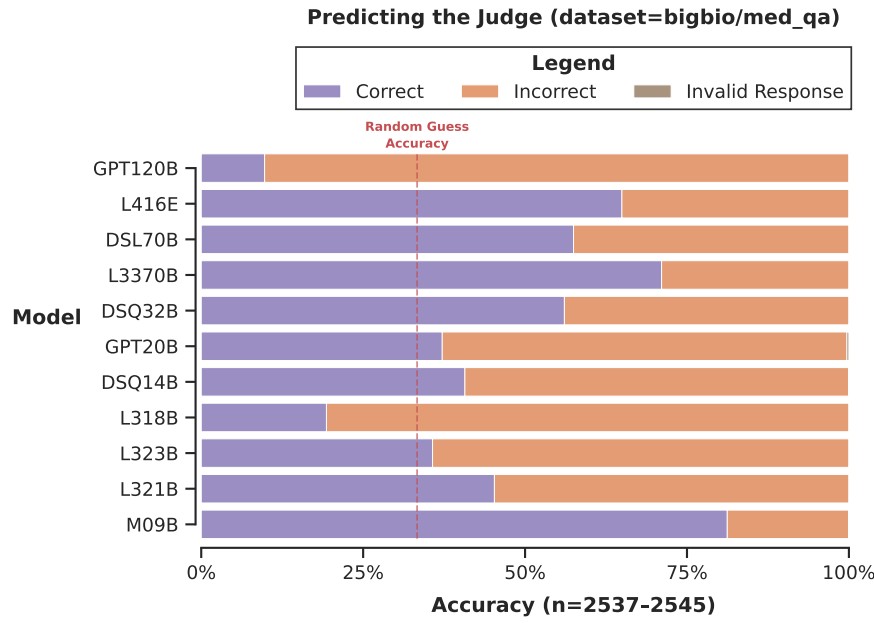

Figure 44: Contextual prediction accuracy of models on MedQA dataset when evaluation is performed according to the mischievous rubric given in Prompt 12.

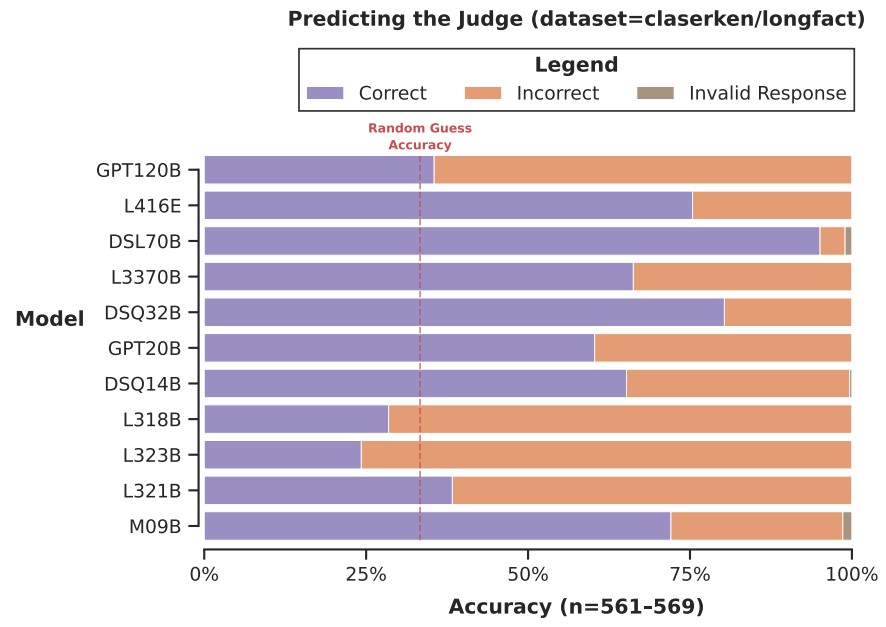

Figure 45: Contextual prediction accuracy of models on LongFact dataset when evaluation is performed according to the mischievous rubric given in Prompt 12.

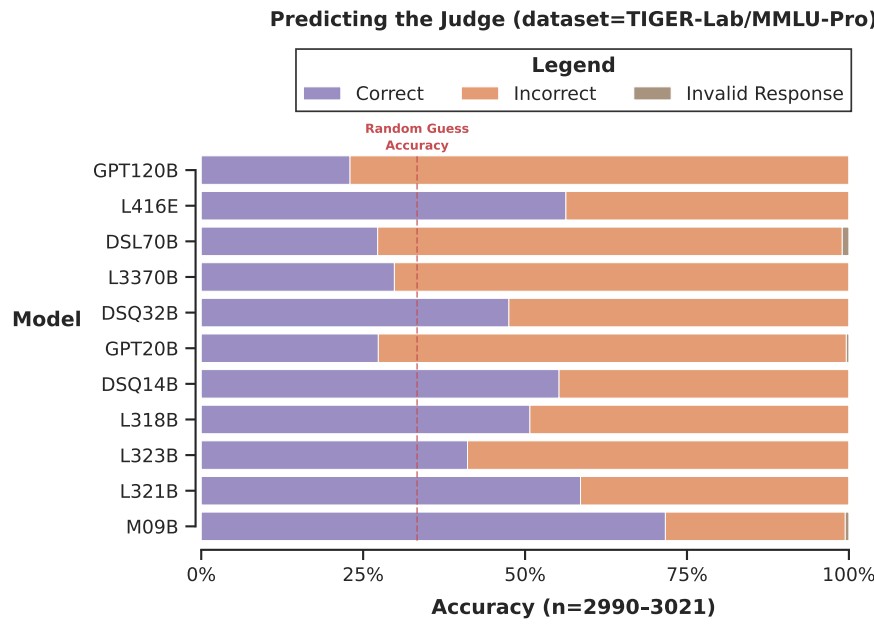

Figure 46: Contextual prediction accuracy of models on MMLU-Pro dataset when evaluation is performed according to the mischievous rubric given in Prompt 12.

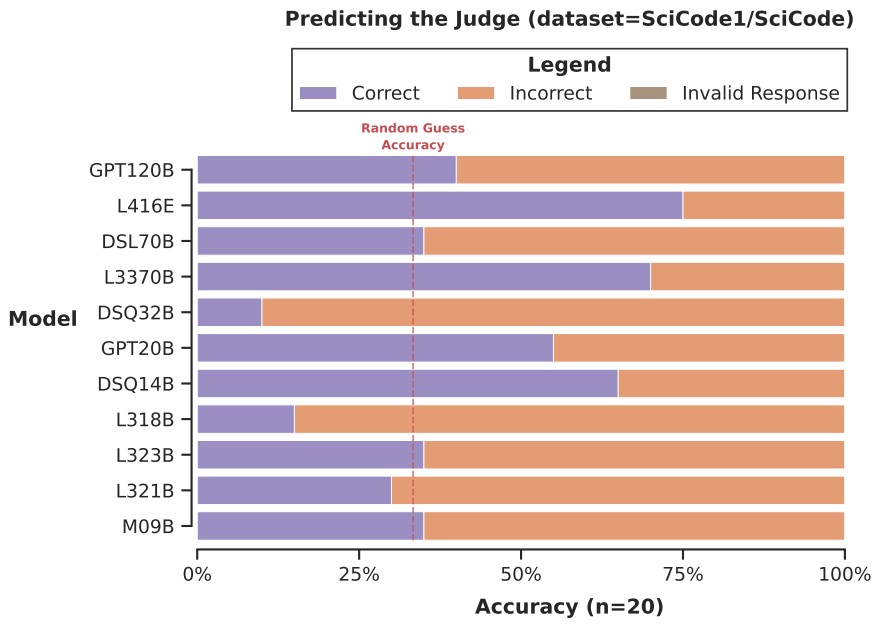

Figure 47: Contextual prediction accuracy of models on SciCode dataset when evaluation is performed according to the mischievous rubric given in Prompt 12.

## J    PREDICTION ACCURACY WITHOUT FINE-TUNED MODELS

As the evaluation is contextual, the results shown in Sections 4 and 5 occur in the context of the fine-tuned models. Figures 48 to 52 show both the zero-shot and contextual prediction accuracy of the models when the fine-tuned models are excluded from the experiment. Evidently, the effect of the fine-tuned models being added to the mix are relatively minimal here.

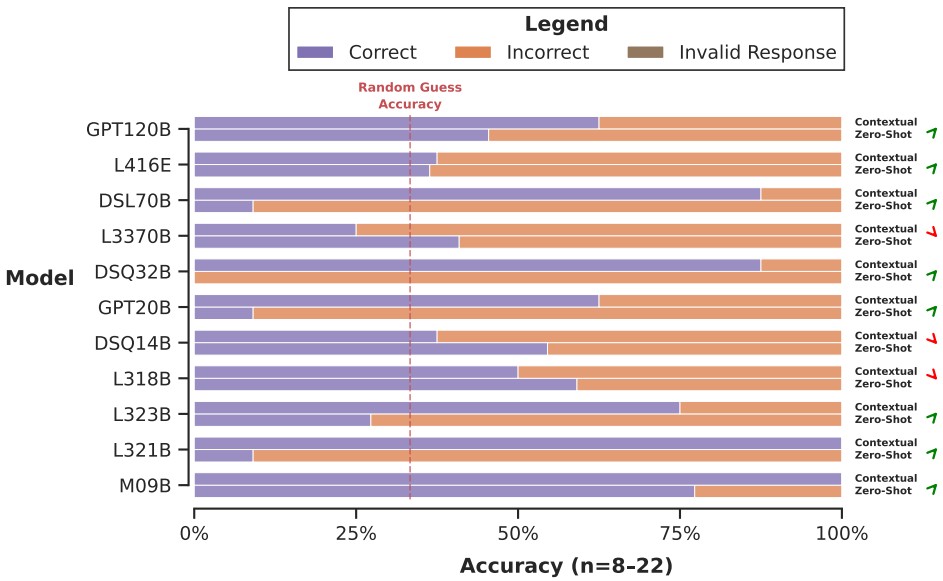

Figure 48: The zero-shot and contextual prediction accuracy of the models on the AIME 2024 dataset.

Figure 49: The zero-shot and contextual prediction accuracy of the models on the MedQA dataset.

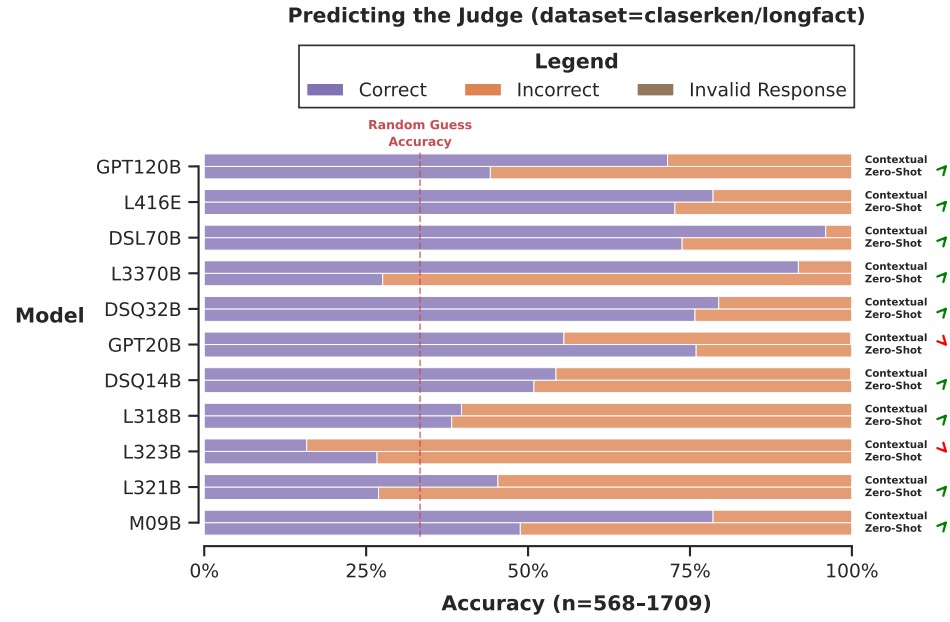

Figure 50: The zero-shot and contextual prediction accuracy of the models on the LongFact dataset.

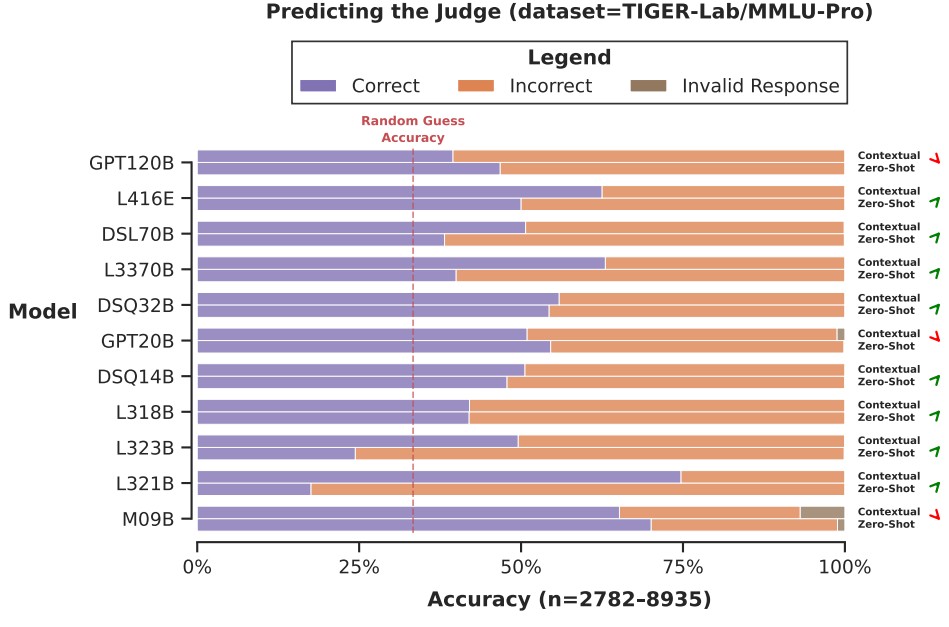

Figure 51: The zero-shot and contextual prediction accuracy of the models on the MMLU-Pro dataset.

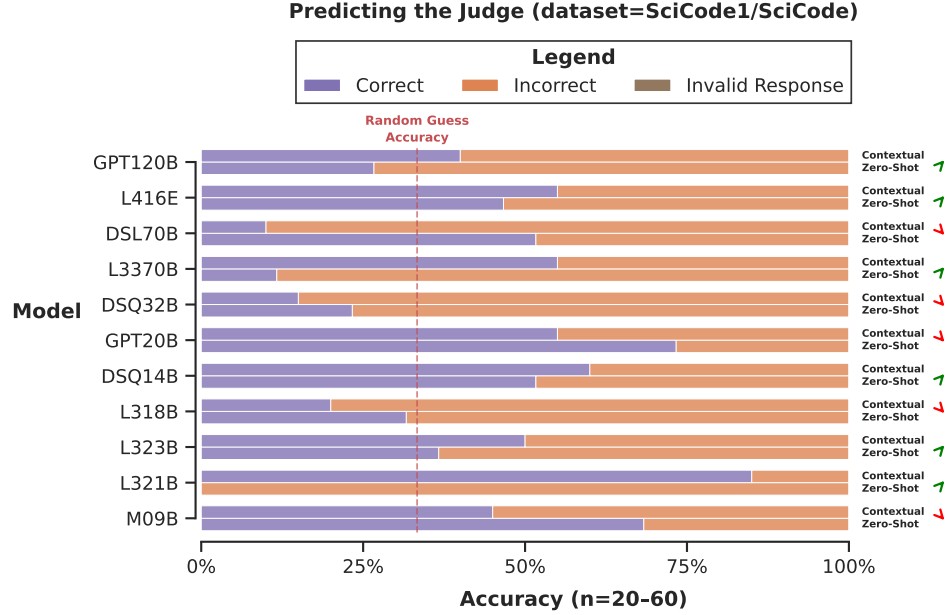

Figure 52: The zero-shot and contextual prediction accuracy of the models on the SciCode dataset.

# K USE OF LARGE LANGUAGE MODELS IN WRITING

LLMs were used throughout the writing of the paper as general-purpose assist tools. In particular, they were used to draft, refine, and polish sections of the paper as well as to discover and compare some relevant works. Only some appendix sections can be said to be exempt of the above, with the contribution of the LLMs in writing being less pronounced around precise factual areas of the text (i.e., where the exact choice of words was critical for correctness).

