# OpenReview forum: "Know When to Fold 'Em: Predicting an LLM-Judge for Efficient but Performant Inference"
_ICLR.cc/2026/Conference — Submitted to ICLR 2026_

### Official Review · Reviewer_nGf2 · 2025-10-26

**Soundness:** 2
**Presentation:** 2
**Contribution:** 2
**Rating:** 2
**Confidence:** 4

**Summary:**

This is a well-executed paper that tackles the practical challenge of balancing an LLM's computational efficiency with its output quality. The core idea is to empower smaller, faster models to "know what they don't know" by having them predict, before generating a full response, how an LLM-judge would score their potential answer. If the predicted score is poor, the query can be deferred to a larger, more capable model. The paper compellingly evaluates three methods for this pre-hoc prediction: zero-shot, an in-context "report card," and supervised fine-tuning. The authors demonstrate that while large models have some innate self-assessment ability, smaller models do not. However, both the report card and fine-tuning methods dramatically improve this capability, with fine-tuning showing the most promise.

**Strengths:**

- The paper addresses a core, real-world problem. The trade-off between cost/latency and quality is a primary concern for anyone deploying LLMs, and the proposed "deferral" system is an elegant solution.

- The concept of an in-context "report card" is a particularly clever, training-free approach. It’s a great interim solution for closed-weight models where fine-tuning isn't an option.

**Weaknesses:**

- The study relies on a single LLM-judge (Llama 3.3 70B). While the judge's evaluations were confirmed to be stable, it would be great to see how the prediction models hold up against different judges (e.g., GPT-4o, Claude 3.5 Sonnet, or even a human panel). Would a model fine-tuned to predict a Llama-judge also be able to predict a GPT-judge?

- The SFT approach is clear, but it would be helpful to have a more detailed discussion on the cost of creating this fine-tuning dataset. It requires generating responses from all models and then running the expensive judge model on them. A brief analysis of this "setup cost" would make the SFT method's practicality even clearer.

- The full, practical implementation of the deferral system isn't explored. It would be fantastic if you could add experiments showing the actual end-to-end performance. For example, a "small model + SFT predictor + large model" system vs. just using the large model, showing the blended cost-per-query and overall quality score.

- The system seems to imply a binary "answer or defer" choice based on "great/ok/bad." It would be interesting to explore a more granular system. For instance, could a predicted "ok" score trigger a simpler, cheaper intervention (like a RAG query) rather than a full deferral to the most expensive model?

-  For the in-context report card method, it would be great to see a more direct analysis of the token overhead vs. the accuracy gain. How many tokens does the report card add to the context, what's the added latency from that, and what is the "break-even" point where the time saved by not generating a bad answer equals the time spent processing the report card?

- There are red "REDACTED" comments at the end of sections like acknowledgement or authors contribution

**Questions:**

- Your judge rubric is quite general. Did you experiment with how prediction accuracy changes if the rubric is made more specific or complex? For instance, if the judge was asked to only score for "factual accuracy" and ignore tone.

- Beyond just prediction accuracy (correctly guessing "great," "ok," or "bad"), did you look at the model's calibration? For example, when the SFT model predicts "bad" with high confidence, is it almost always correct?

---

> ### Author Response · Authors · 2025-12-03
> **Rebuttal**
>
> We thank the reviewer for their time and attention in reviewing our paper.
>
> **W1)** This weakness relates specifically to how similar LLM judges are in their evaluations and so is outside the scope of this work. Results from some of the cited papers (e.g., Agent-as-a-Judge) confirm that these judges align with human responses, implying this would hold for any reasonable judges without additional experiments. We have now conducted additional experiments with a separate judge model (GPT OSS 120B), which confirms that the approach carries over to different judges. While this does not directly address your comment, we note that the combination of this result and prior work that shows the alignment between judges implies that the prediction accuracy would be similar if equally powerful judges were used interchangeably in training and testing.
>
> **W2)** This dataset is produced naturally in the zero-shot phase of our experiments and requires no additional overhead. It is collapsed into the joint cost of all the experiments reported in the reproducibility statement.
>
> **W3)** We agree with the reviewer that this is a key point not addressed here. However, the question of how this should be best integrated into a system, and the complexities thereof, is both too space-consuming to be included here (and not suitable for an appendix) and outside the scope of this work. Direct follow-up work will address this, but proof of the ability to perform the prediction must precede such work.
>
> **W4)** Indeed, such a thing is possible. However, building a more complicated routing system must follow work showing that prediction is possible in the first place. Thus, we leave it as future work, but are already planning to address this in follow-up work.
>
> **W5)** The number of additional input tokens (note that processing input tokens is typically 10x--100x faster than output tokens) is relatively small, and inference time on a small model is low. Thus, the actual additional overhead with a small model predicting its score first is negligible compared with directly performing response generation on a big model. Thus, when visualizing this, we get a relatively uninteresting graph which is not worth including in the work.
>
> **W6)** That is due to the double-blind policy. We will be replacing them (alongside the source code one) with actual text if accepted.
>
> **Q1)** We have now run an ablation experiment on whether a different rubric is used, whether the model can still predict the responses (Appendix I). The result shows that prediction accuracy drops (still remaining significantly better than random) when the report cards are used for a contrarian rubric. This suggests that there is some transfer between zero-shot and contextual performance (and, by extension, between reasonable judges).
>
> **Q2)** We include prediction distribution results in Appendix F. In general, the predictions of the model correspond most closely to the dataset a query comes from, suggesting that the fine-tuning is serving a similar purpose to the report cards. Additional improvements on the data mixing in the training phase would likely reduce the frequency of this.
>
> Once again, we thank the reviewer for their time and attention in reviewing the paper. We understand that the situation with the hack prevents further discussion here, but we hope we have been able to allay your most pressing concerns.

---

### Official Review · Reviewer_f7ts · 2025-10-30

**Soundness:** 3
**Presentation:** 2
**Contribution:** 2
**Rating:** 4
**Confidence:** 3

**Summary:**

This paper explores whether LLMs can anticipate how an external LLM-based judge would evaluate their answers before actually generating them. The authors investigate three strategies: (1) zero-shot prediction, (2) in-context “report card” prompting that summarizes past performance, and (3) supervised fine-tuning using the hindsight trick. Results show that larger reasoning models display reasonable self-assessment ability even in zero-shot settings, while smaller models benefit significantly from contextual report cards or fine-tuning.

**Strengths:**

1. The paper addresses how to make LLMs self-aware enough to know when to ask for help. This is both conceptually interesting and relevant for efficient LLM deployment.

2. The proposed approaches are well-motivated and systematically compared across diverse datasets and model sizes.

3. The authors provide some detailed empirical results and ablations (e.g., per-category analysis on MMLU-Pro) that reinforce their claims.

**Weaknesses:**

1. All experiments rely on a single LLM judge (Llama-70B). This raises questions about generalization to different evaluators or judging paradigms.

2. Since both judge and fine-tuning signals ultimately depend on LLM-generated labels, there is no ground truth accuracy. Also, the number of classification types is small (only 3), making the classification problem seem easy.

3. The report cards are generated on a training set, and then given to the testing set as part of the prompt. As this paper considers the i.i.d. case testing, this may give too much shortcut for the problem, making it hard to tell how much improvement is coming aside from just aligning the distribution of the test and training set.

**Questions:**

See Weaknesses.

---

> ### Author Response · Authors · 2025-12-03
> **Rebuttal**
>
> We thank the reviewer for their time and attention in reviewing our paper.
>
> **1)** We explicitly note this as a limitation in our work. However, we are confused by this comment. The strategies we employ should naturally extend to any reasonable judge, working in direct proportion to the judge's strength and predictability. A more sophisticated judge is expected to be as easy to predict, or easier, than the one we use. To further address this, we have now run an ablation experiment using GPT OSS 120B as a judge (Appendix G). Our results show that the above assumptions hold.
>
> **2)** The ground truth is the LLM judge's response. There is no other possible ground truth here, as there are many answers that would be suitable responses to a query, making the absence of the kind of ground truth the reviewer is referring to a challenge of the problem setting. As we note in the work, it is established that reasonable LLM judges are strongly correlated with ground truths in other datasets, implying that they can serve the same purpose here.
>
> **3)** We agree with the reviewer that this is a weakness, and we thank them for pointing it out. However, this is naturally bypassed by simply having more datasets in the report cards or fine-tuning data (an extensive run over many datasets would be the case in deployment of such a system; note that our results on the shorter report card show that more information helps make more accurate predictions). Thus, we do not feel that the inclusion of such a result would serve to bolster the strength of the paper.
>
> Once again, we thank the reviewer for their time and attention in reviewing the paper. We understand that the situation with the hack prevents further discussion here, but we hope we have been able to allay your most pressing concerns.

---

### Official Review · Reviewer_ycqw · 2025-10-31

**Soundness:** 3
**Presentation:** 2
**Contribution:** 2
**Rating:** 2
**Confidence:** 4

**Summary:**

The paper introduces a pre-self-assessment mechanism that enables large language models (LLMs) to predict, prior to generation, how an LLM-based judge would evaluate their responses. It explores three strategies—zero-shot, in-context report cards, and supervised fine-tuning—and demonstrates that smaller models can be effectively calibrated to route queries to larger models only when needed, achieving efficient inference under resource constraints.

**Strengths:**

1. Methodological novelty

The report card approach is innovative. It consolidates joint judge evaluations across multiple models and datasets into concise textual performance summaries, eliminating the need for expensive per-query judge calls. By combining hindsight relabeling with supervised fine-tuning (SFT), the method effectively repurposes existing judge scores as supervision signals. An ablation study (Appendix E) comparing joint versus isolated judging further shows that joint evaluation enhances score diversity.

2. Broad & reproducible evaluation

The study presents extensive experiments across five diverse datasets (MedQA, LongFact, AIME’24, SciCode, MMLU-Pro) and eleven models ranging from 0.9B to 120B parameters, including recent reasoning architectures such as Llama-4 Scout and DeepSeek-R1. The release of complete prompts (Appendix C) and the detailed judge rubric reinforces the work’s reproducibility and transparency.

3. Strong empirical findings

Fine-tuning yields substantial improvements, with the some model achieving a +52 percentage point gain in prediction accuracy. Even large non-reasoning models benefit from contextual report cards. Notably, prediction accuracy increases with query difficulty, suggesting that task complexity itself can serve as an informative signal for adaptive model routing.

**Weaknesses:**

1. Reliance on a Single Judge Model

The study depends exclusively on a single LLM judge (Llama 3.3 70B) for all evaluations, raising concerns about evaluation bias and potential overfitting to one model’s judgment criteria. Without comparisons across multiple judges or human evaluations, the generality and robustness of the proposed approach remain uncertain.

2. Methodological Simplicity and Incomplete Framework

The three proposed methods—zero-shot probability prediction, contextual report card prompting, and supervised fine-tuning—are conceptually straightforward, relying on techniques such as prompt engineering and fine-tuning that have been widely explored in prior work. Moreover, the paper does not clearly specify how queries predicted as “bad” are routed to larger models, leaving the proposed self-assessment-based routing framework incomplete for real-world deployment.

3. Unrealistic Assumption of Report Card Availability

The report card method assumes access to detailed historical performance summaries for each model across multiple datasets. In practical settings, such comprehensive records are rarely available, particularly for unseen data. This assumption limits the method’s applicability and generalization potential.

4. Lack of Baselines and Comparative Analysis

The paper omits key baselines such as uncertainty modeling and self-evaluation approaches. Without these comparisons, it is difficult to assess the relative improvement, effectiveness, or novelty of the proposed techniques.

**Questions:**

1. The paper discusses agent in both PRELIMINARIES and RELATED WORK, but the main methods and contents of the paper do not seem to involve agent?

2. The PRELIMINARIES section contains some redundant details. The description of LLM architectures is not directly relevant to the paper’s central research question—predicting LLM judge scoring. It may be beneficial to simplify this section and focus only on the components essential to understanding the proposed methods.

---

> ### Author Response · Authors · 2025-12-03
> **Rebuttal**
>
> We thank the reviewer for their time and attention in reviewing our paper.
>
> **W1)** We are confused about this comment. The goal is to overfit to a judge model, as the judge model's preference determines the optimization goal. However, we have now run an experiment showing that these results are consistent if the judge model is replaced by a different one (GPT OSS 120B). These results are presented in Appendix G. Regarding alignment with human evaluations, we already know this from prior work that examined the alignment between such judges and human evaluators (e.g., Agent-as-a-Judge).
>
> **W2)** We again remain somewhat confused by this comment. Methodological simplicity is a strength of this work. The problem is important, and if the solution is simple, then that is a good thing. Regarding basic routing behaviour, the most basic setting is clearly shown in Figure 2. More complicated deployment settings are left as future work, which must first be preceded by work showing such predictions are possible (i.e., this work).
>
> **W3)** We continue to be confused. The performance of a model on different datasets is one of the most commonly available properties of a model. Most models come with evaluation performance across many datasets (allowing for variants of this setting to be directly used), and producing a report card for a model in the way presented here can be done by running a model (with either open or closed weights) for at most a few hundred queries. This is incredibly inexpensive compared to most contemporary work involving LLMs.
>
> **W4)** Once more, we are confused by the reviewer's comment. There are no suitable baselines here below zero-shot evaluation. We are proposing a problem here with broad utility and showing that simple methods can solve the problem. Uncertainty modelling or self-evaluation is unrelated to the problem.
>
> **Q1)** The inclusion of an LLM judge renders this an agentic system (see Figure 1), and the application of this work would also be within an agentic system (see Figure 2).
>
> **Q2)** We respectfully disagree. It is strange to imagine writing a paper involving predicting an LLM judge without actually explaining what an LLM is. We only include the most high-level key details of an LLM here.
>
> Once again, we thank the reviewer for their time and attention in reviewing the paper. We understand that the situation with the hack prevents further discussion here, but we hope we have been able to allay your most pressing concerns.

---

### Official Review · Reviewer_rVge · 2025-10-31

**Soundness:** 3
**Presentation:** 3
**Contribution:** 3
**Rating:** 2
**Confidence:** 3

**Summary:**

This paper investigates whether language models can predict LLM judge scores before generating responses, enabling efficient routing where small models handle easy queries and defer hard ones to larger models. This is practically relevant for 2025's on-device AI deployment trends.
The study tests three approaches across 11 models and 5 datasets: zero-shot prediction, report cards with historical performance, and fine-tuning. Key finding: reasoning models show inherent self-awareness while small models achieve up to 52% improvement with fine-tuning.

**Strengths:**

1. Timely practical problem addressing real deployment challenges with comprehensive experiments across medical, mathematical, coding, and factual domains.
2. Strong empirical findings demonstrating that small models can learn their limitations, providing actionable baselines for production systems.
3. Well-documented reproducible methodology with extensive ablations.

**Weaknesses:**

1. Zero technical innovation: zero-shot is basic prompting, report cards are standard in-context learning, fine-tuning is vanilla supervised learning from 2022.
2. All tasks have objective answers (multiple choice, math solutions, code correctness). No subjective tasks like creative writing or advice-giving where evaluation is ambiguous.
3. Single judge (Llama 3.3 70B) creates uncertainty whether models learn true self-awareness or just memorize one judge's preferences.

Critical Questions

- For LongFact success: what drives correct predictions? Is it response length estimation, keyword matching, or topic familiarity? Need 10-20 concrete examples showing why model correctly predicted "great" versus "bad" with feature analysis.
- For AIME math problems: how do models assess difficulty? Is prediction based on problem length, mathematical terminology, or actual computational complexity? Requires stratified analysis by difficulty level showing models correctly identify when they fail on olympiad problems but succeed on algebra.
- Which evaluation criteria drive predictions? The rubric includes accuracy, relevance, clarity, formatting. Do models fail when answers are correct but poorly formatted? When verbose but accurate? Need ablation isolating each criterion's influence.

**Questions:**

- RAG integration: does retrieval-augmented generation improve prediction accuracy on knowledge-intensive queries by providing reference context?
- Federated learning: can distributed edge devices collaboratively build report cards without sharing raw data, learning when to defer to cloud models based on collective experience?
- Cross-judge generalization: train on Judge A, test on Judge B and human ratings to distinguish true self-awareness from judge-specific overfitting.

---

> ### Author Response · Authors · 2025-12-03
> **Rebuttal**
>
> We thank the reviewer for their time and attention in reviewing our paper.
>
> **W1)** The technical innovation comes from the problem and the approach. The simplicity of the solution presented here is a strength of the paper, and not a weakness. The fact that vanilla SFT is sufficient is a more useful result to the community than introducing a more complicated method would be.
>
> **W2)** We are somewhat confused by this comment, as the reviewer's comment is a bit inaccurate. Datasets like LongFact do NOT have a set of correct responses, and the rubric covers aspects that would be present in creative writing, such as the clarity of the language. Thus, this comment is more about whether an LLM judge can accurately gauge the quality of a bit of writing and not whether it is possible to predict a response given by them. This comment is, thus, regarding a topic outside the scope of this work.
>
> **W3)** We believe that the reviewer may have misunderstood some key aspects of the paper. Memorizing a judge's preference is the goal here; self-awareness is in the context of the judge's preference. We have now conducted both an ablation with GPT OSS 120B as the judge (Appendix G) and with a different rubric (Appendix I). Our results suggest that the approach can generalize to different judges and preferences, but that models are better when the judge is more reasonable (as would be expected).
>
> **CQ1)** This is orthogonal to the objective of the work here. The goal here is to have the model predict the response based on a history of answering similar questions. The models rely on the abstract categorization of the query to predict their performance. Performing a more detailed analysis would thus increase bloat while not supporting the core objective of the work here.
>
> **CQ2)** This is, again, orthogonal to the objective of the work here. The goal here is to have the model predict the response based on a history of answering similar questions. Given a higher granularity of maths questions, the categorization the models rely on would become more granular; if we only use one kind of math question in the report cards, they only have their performance on one kind of math question to gauge how they will perform on all maths problems (and thus will treat all maths problems the same). Performing a more detailed analysis would increase bloat while not supporting the core objective of the work here.
>
> **CQ3)** Once more, this is orthogonal to the objectives of the work here. Why the models give their predictions is irrelevant to whether they can predict the judge. This work is a first step that shows that these models can predict their own performance. Improving on that prediction accuracy by more deeply understanding why they make predictions is a different topic that must necessarily follow this one, and so must remain future work.
>
> **Q1)** This is naturally the case. While we don't explore using more sophisticated agentic systems for the model here, naturally, these would boost the performance. We leave the implementation details of this within a full agentic system to future work.
>
> **Q2)** We thank the reviewer for this exceptionally good point. Indeed, this is trivially possible, as this work shows that collapsing the performance of a model down to even just modes is enough to predict performance.
>
> **Q3)** The goal here is to predict how the models will score against a specific judge. This question relates specifically to how similar LLM judges are in their evaluations and so is outside the scope of this work. Results from some of the cited papers (e.g., Agent-as-a-Judge) confirm that these judges do align with human responses, implying that this would carry over without necessitating additional experiments to show it. We have conducted additional experiments with a separate judge model (GPT OSS 120B), which confirms the approach carries over to different judges. While this does not directly address your comment, we note that the combination of this result and prior work that shows the alignment between judges implies that the prediction accuracy would be similar if equally powerful judges were used interchangeably in training and testing.
>
> Once again, we thank the reviewer for their time and attention in reviewing the paper. We understand that the situation with the hack prevents further discussion here, but we hope we have been able to allay your most pressing concerns.

---

### Author Response · Authors · 2025-12-03
**Rebuttals**

We thank all the reviewers for their time and attention in reviewing our paper. We have provided a rebuttal for each review. We understand that the situation with the hack prevents further discussion here, but we hope we have been able to allay all of the most pressing concerns presented here.

---

### Meta-Review · Area_Chair_hxxG · 2025-12-03

**Summary:**

This paper proposes to predict the score a judge would assign to a LLM’s outputs prior to generating those outputs.  If successful, this would be useful for deciding which model is the cheapest model that would successfully answer a query, improving efficiency.  The paper’s empirical studies show that large models are already capable of this behavior whereas small models require some finetuning or in-context report cards.  Reviewers raised the following significant concerns: (1) all techniques used are classic existing techniques so there is no novelty in methodology, (2) didn’t explore subjective tasks without clear right and wrong answers, (3) single judge used, (4) needs finegrained analysis on what drives correct predictions, (5) paper does not specify how they would use their techniques in a real deployment for the purpose of routing as suggested by their motivation, (6) lack of baselines, (7) paper should include a study of latency where they compare the cost of the report cards to the savings.

**Reviewer Concerns:**

No points were addressed since no rebuttals were posted.

**Reviewer Scores:**

The original scores were 2, 2, 4, 2.  Since the authors did not post rebuttals, I assume the reviewers would not change their scores

---

### Decision · Program_Chairs · 2026-01-26

Reject